# Enhancing Personalized Multi-Turn Dialogue with Curiosity Reward

**Yanming Wan**[2][*][†][‡]**, Jiaxing Wu**[1][*][†]**, Marwa Abdulhai**[4]**, Lior Shani**[3]**, Natasha Jaques**[12]

[1]Google DeepMind [2]University of Washington
[3]Google Research [4]University of California, Berkeley

[*]Equal Contribution [‡]Work done during internship at Google DeepMind
[†]Correspondence to: <ymwan@cs.washington.edu, jxwu@google.com>

## Abstract

Effective conversational agents like large language models (LLMs) must personalize their interactions to adapt to user preferences, personalities, and attributes across diverse domains like education and healthcare. Current methods like Reinforcement Learning from Human Feedback (RLHF), often prioritize helpfulness and safety but fall short in fostering truly empathetic, adaptive, and personalized dialogues. Existing personalization approaches typically rely on extensive user history, limiting their effectiveness for new or context-limited users. To address these limitations, we propose leveraging a user model to incorporate a curiosity-based intrinsic reward into multi-turn RLHF. This novel reward mechanism encourages the LLM agent to actively infer user traits by optimizing conversations to improve its user model's accuracy. Consequently, the agent delivers more personalized interactions by learning more about the user. We demonstrate our method's effectiveness in two distinct domains: significantly improving personalization performance in a conversational recommendation task, and personalizing conversations for different learning styles in an educational setting. We show improved generalization capabilities compared to traditional multi-turn RLHF, all while maintaining conversation quality. Our method offers a promising solution for creating more personalized, adaptive, and engaging conversational agents.

## 1 Introduction

Deploying large language models (LLMs) in open-ended conversations requires more than just generic responses—it demands adaptation to each user's unique context, including their needs, goals, personality, and evolving preferences. An effective conversational agent should feel like a personalized companion, tailoring its answers, writing style, and tone as it learns about the individual. This level of personalization is especially crucial in human-centric applications such as education and healthcare, where one size does not fit all. However, current training paradigms for LLMs, including reinforcement learning from human feedback (RLHF), fall short of this goal. They typically rely on a single unified reward function applied uniformly across users, and optimize in single-turn interactions, **ignoring long-term personalization**. As a result, conventional RLHF-trained models tend to average over user preferences, failing to account for individual differences [1, 2].

Personalization is not just a luxury but often a necessity for effectiveness. In educational settings, adaptive teaching methods that respond to a learner's knowledge level and learning style can dramatically improve engagement and outcomes [3]. Similarly, in therapeutic contexts, a conversation agent must be sensitive to a user's emotional state and personal history, adjusting its interactions to build

39th Conference on Neural Information Processing Systems (NeurIPS 2025).

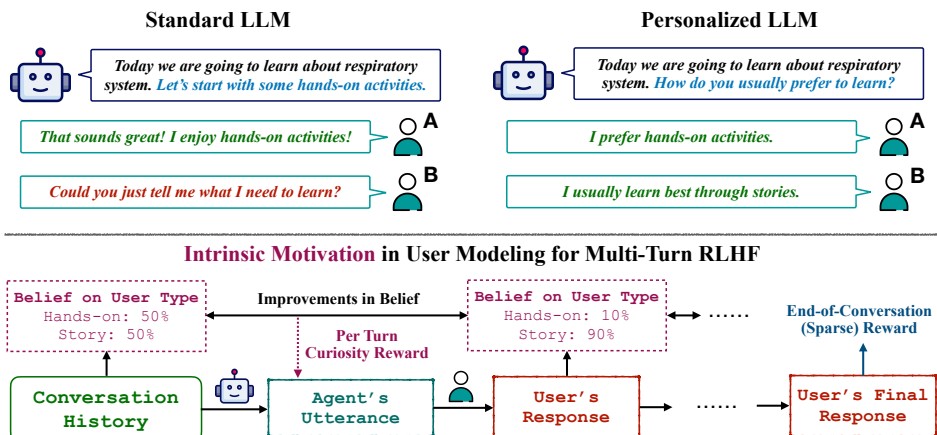

Figure 1: Our work focuses on training personalized LLMs in multi-turn conversations. Standard LLM training methods treat all users as a homogeneous group, leading to suboptimal performance for different groups (top left); while an optimal LLM can actively learn about user preferences within the conversation and then adapt to it (top right). We introduce Intrinsic Motivation in user modeling to multi-turn RLHF. Intuitively, rather than training an LLM only with the end-of-conversation sparse reward, we add an additional turn-based reward that is given by its improvement in belief over the user type after generating an utterance and receiving a response, which guides the LLM to actively learn about user type and then adapt to each user throughout the conversation.

trust and efficacy [4]. Intuitively, tailoring interactions to the individual can enhance user satisfaction, engagement, and overall success of the intervention, which indicates that an LLM that dynamically personalizes its behavior holds immense promise for improving user experience and effectiveness in a range of applications. Despite this importance, most existing approaches to personalize LLMs require extensive pre-collected user data or profiles. Recent works on aligning models to user-specific preferences often assume access to a user profile, history, or latent representation gathered prior to the conversation [5, 6, 7, 8, 9]. For example, reward-modeling techniques have been proposed to infer latent user clusters or employ user-specific fine-tuning, but these typically involve additional training on feedback data from each user ahead of time [5]. Such requirements limit the practicality of personalization: in real-world deployments, we may not have rich user data in advance. This gap motivates us to develop methods for **online personalization, where the LLM learns about the user during the conversation**, reducing its uncertainty about the user's traits as the dialogue unfolds.

In this paper, we propose a novel method to enhance LLMs' ability to conduct personalized multi-turn conversations, which we call **C**uriosity-driven **U**ser-modeling **R**eward as an **I**ntrinsic **O**bjective (**CURIO**). Intrinsic motivation, particularly curiosity, has a rich history in traditional reinforcement learning (RL) frameworks (e.g., [10, 11, 12, 13, 14, 15]). However, these approaches have not yet been adapted to the rapidly evolving domain of LLM post-training, primarily due to the significant computational complexity involved in maintaining both an LLM policy and a separate environment model simultaneously. We are the first to bridge this gap by incorporating intrinsic motivation into the LLM paradigm through an intrinsic reward mechanism. Typically, in RL applications curiosity is used to reduce uncertainty in a model of the world. Our key insight is that in dialog, the user is the world in which the agent interacts. Therefore, we learn a model of the user, and implement a curiosity objective that is designed to conduct conversations to increase the accuracy of our user model. Intuitively, this strategy promotes a dynamic balance between exploiting conversation rewards, and exploring to learn more about the environment (the users). This enables the model to adapt its interaction style for effective personalized conversations. To facilitate this, we have developed an advanced engineering framework capable of orchestrating multiple LLMs, efficiently supporting online multi-turn RL policy updates. Theoretically, personalized conversation can be formulated as a Partially Observable Markov Decision Process (POMDP) with belief-state updates. We show that our intrinsic reward can be designed as a Potential-based Reward Shaping (PBRS) [16] approach that doesn't change the optimal policy [17] but enhances sample efficiency when properly implemented, showing for the first time how PBRS can be applied to train LLMs.

Figure 1 illustrates our approach: the LLM receives intrinsic rewards based on improvements in its belief about the user after each conversational turn. This turn-based reward complements the sparse end-of-conversation reward. By leveraging multi-turn RL combined with such intrinsic rewards, our model learns to strategically plan actions that facilitate continuous learning about the user throughout the conversation. Consequently, it progressively refines its understanding, effectively **learning how to learn about the user**.

We empirically evaluate CURIO on two conversational tasks—Education Dialogue [18] and Exercise Recommendation. Given the considerable challenge of applying theoretical RL concepts to practical LLM fine-tuning, we selected these tasks as well-defined and controlled benchmarks. Our experiments clearly demonstrate CURIO's superior performance in rapidly adapting to individual users. Specifically, CURIO motivates the LLM to actively reduce uncertainty about users by asking insightful questions and generating context-sensitive responses. The generalized learning capabilities of our approach enable the LLM to quickly and effectively personalize interactions even for entirely unseen users, consistently achieving better performance compared to latest multi-turn RLHF baselines.

In summary, our contributions are:

- **A novel framework (CURIO) for personalized dialogue with LLMs:** We reformulate multi-turn RL fine-tuning of LLMs to include personalization, and by leveraging the **user model** we introduce a **curiosity-based intrinsic reward** that drives the policy to learn about and adapt to the user within the conversation.

- **Connection with theoretical results:** We theoretically connect our approach to potential-based reward shaping, providing a formal justification for our intrinsic reward design.

- **Benchmarking personalization in conversations:** We establish an evaluation protocol over two distinct domains — Education Dialogue and Exercise Recommendation. This protocol assesses an LLM-based conversational agent's capacity to infer user traits and adapt its interactions dynamically within multi-turn dialogues.

- **Enhanced personalization through adaptive learning:** We quantitatively demonstrate that our curiosity-driven approach with auxiliary user modeling significantly outperforms standard multi-turn RLHF in adapting to diverse users and demonstrates better generalization capability, while preserving conversation quality. We further provide qualitative analysis over the performances of baselines and various designs of intrinsic reward.

## 2 Related Works

**Reinforcement Learning in LLMs.** RLHF is widely used for aligning language models with general user preferences [19]. Ouyang et al. [1] trained models using aggregated human judgments, resulting in broadly helpful assistants. However, conventional RLHF methods rely on a universal reward function, neglecting individual user preferences by effectively optimizing for an "average user," leading to suboptimal performance when preferences diverge [2, 5, 7, 9]. To address this limitation, several personalized RLHF approaches have been proposed. Poddar et al. [5] propose to infer a latent context vector for each user, enabling the reward model (and policy) to adjust to that user's revealed preferences. Similarly, Chen et al. [7] learn a latent preference space covering heterogeneous user opinions; their method trains a reward function that can generalize to new users with a few examples by modeling each user as a point in this latent space. Wu et al. [6] extract reward signals from downstream personalization tasks to generate natural language user profiles, which are then used to personalize LLMs. Shenfeld et al. [9] formulate an individual's reward as a weighted sum of base reward functions and uses a small number of preference queries to infer the user-specific weights. These personalized alignment methods indeed tailor an LLM's behavior to different users, but they require additional user-specific info or prep work *before* the personalized interaction can take place.

In contrast, our method does not require any separate calibration or auxiliary user profile in advance. The personalization of the agent emerges dynamically through multi-turn interactions: as the conversation unfolds, the model infers the user's traits and preferences and adapts its responses accordingly. This on-the-fly learning of user preferences means our approach can personalize in real-time without an upfront personalization phase, which is a key differentiator from prior RLHF-based personalization techniques. Hong et al. [20] also propose to leverage the multi-turn setting to learn about the user, but they mainly focus on training offline-RL agents over synthetic data to optimize goal-directed

objectives (explaining concepts or recommend activities). Our agents, however, *explore how to learn user preferences throughout conversations* with online-RL. Such ability to actively infer user preferences during the conversation can bring additional benefits in open-ended dialogues. In the absence of a clearly defined task, the enjoyability of the interaction itself becomes an important consideration. Encouraging users to voluntarily share personal ideas can enhance their engagement and overall enjoyment of the conversation [21, 22, 23], which is not realizable for traditional approaches that primarily focus on helpfulness and harmlessness.

**Personalized Conversation.** Personalized dialogue systems have been extensively studied in domains like education and therapy, demonstrating enhanced learning, adherence, and user satisfaction. Examples include *AutoTutor* [24] for adapting hints, virtual counselors by Bickmore et al. [25] for rapport building, and Woebot for CBT [26]. However, existing personalized agents are typically domain-specific, relying on limited data that hampers generalizability. While recent large language model (LLM) approaches are emerging, they often remain application-specific. In contrast, our method employs a domain-agnostic LLM capable of dynamically inferring user preferences, facilitating personalized interactions that generalize across diverse contexts and populations.

**Intrinsic Motivation.** Our work also connects to research on intrinsic motivation and curiosity-driven learning [12, 13] in reinforcement learning. Intrinsic rewards—bonus signals not directly tied to the task's external goal—have been used extensively to encourage agents to explore novel states or learn useful information. Specifically, VIME [13] gives an agent reward for reducing uncertainty in its dynamics model, effectively rewarding information gain about the environment. Such methods have proven effective in complex environments with sparse external feedback, as they drive the agent to discover new states and behaviors by itself. The intrinsic reward can be seen as a form of reward shaping. In reinforcement learning theory, adding a shaping reward (derived from a potential function over states) does not alter the optimal policy, but can accelerate exploration and learning [16, 17]. Recently, Lidayan et al. [27] explicitly links the intrinsic motivation to reward shaping through a theoretical framework, but their empirical analysis is limited to some simple RL domains. We bring the classical concept of intrinsic motivation into the LLM domain by designing intrinsic rewards that guide the model to actively ask insightful questions and explore user attributes in multi-turn dialogue. To our knowledge, this is the first application of such techniques in the LLM domain.

## 3 Curiosity-driven User-modeling Reward as Intrinsic Objective (CURIO)

**Preliminaries.** In traditional RLHF, a conversational task is commonly formulated as a Markov Decision Process (MDP), defined by the tuple $(\mathcal{S}, \mathcal{A}, \mathcal{T}, \mathcal{R}, \gamma)$. At time step $t$, the state $s_t \in \mathcal{S}$ represents the current conversation rollout, and the action $a_t \in \mathcal{A}$ is the response generated by our language model. The user then contributes to generate the next state $s_{t+1}$ by appending action $a_t$ and their corresponding utterance to the current state $s_t$. The transition dynamics $\mathcal{T} : \mathcal{S} \times \mathcal{A} \rightarrow \Delta(\mathcal{S})$ defines the distribution over the next state given the current state and action, and $\mathcal{R} : \mathcal{S} \times \mathcal{A} \rightarrow \mathbb{R}$ denotes the reward function evaluating the quality of each action. The agent aims to optimize the expected cumulative reward, represented by the value function $V^\pi(s_0) = \mathbb{E}\left[\sum_{t=0}^\infty \gamma^t \mathcal{R}(s_t, a_t) \mid \pi\right]$ where $\pi : \mathcal{S} \rightarrow \mathcal{A}$ is the policy, and $\gamma \in [0, 1)$ is the discount factor. The expectation is taken over $a_t \sim \pi(\cdot \mid s_t)$ and $s_{t+1} \sim \mathcal{T}(\cdot \mid s_t, a_t)$.

### 3.1 Personalization as User-Conditioned RLHF

To extend this formulation to personalized conversational tasks, we introduce the user type $u \in \mathcal{U}$, which we assume is fixed throughout the conversation. For each user $u$, the transition dynamics and reward function are conditioned on $u$, meaning that different users may respond differently and provide different preference ratings. However, the user type is unobservable in most real world settings. On one hand, extensive user background information is usually not accessible to LLMs *a priori*. On the other hand, when LLMs are trained on a large corpus collected by annotators from all over the world, it is inherently learning a mixture of unknown diverse users.

Consequently, the problem can be modeled as a Partially Observable Markov Decision Process (POMDP), defined by the tuple $(\tilde{\mathcal{S}}, \mathcal{U}, \mathcal{A}, \tilde{\mathcal{T}}, \tilde{\mathcal{R}}, \gamma)$. Specifically, we define $\tilde{s}_t = \langle s_t, u \rangle$ to be the *"extended"* states in the POMDP, where $s_t$ is still observable but $u$ is unobservable. The transition dynamics and the reward function are defined over the extended states, and thus conditioned on the user type. Formally, we have $\tilde{\mathcal{T}}(\tilde{s}_{t+1} \mid \tilde{s}_t, a_t) = \mathcal{T}(s_{t+1} \mid s_t, a_t, u)$ and $\tilde{\mathcal{R}}(\tilde{s}_t, a_t) = \mathcal{R}(s_t, a_t \mid u)$.

Now we consider an LLM agent in this POMDP environment. Although it does not know the ground truth user type initially, it can maintain a belief over the user type and update its belief as it receives more responses from the user. Therefore, we define the belief function at time step $t$ as $b_t \in \Delta(\mathcal{U})$, which is a probability distribution over all possible user types. If the agent has an initial belief $b_0$, then a Bayesian belief update is formulated as:

$$b_{t+1}(u) \propto \mathcal{T}(s_{t+1} \mid s_t, a_t, u) b_t(u). \tag{1}$$

Note that in the language setting, $s_{t+1}$ contains the concatenation of $s_t$ and $a_t$ (the previous conversation history and the next response), so we can define the belief function as $b_{t+1} = f_{b_0}(s_{t+1})$ based on this recursive relation. In real settings, $f_{b_0}$ can be any belief function $\mathcal{S} \to \Delta(\mathcal{U})$ given that the belief update might be sub-optimal. Since the agent has uncertain beliefs over the true user type, it commonly computes the expected rewards over the belief distribution:

$$\mathcal{R}^b(s_t, b_t, a_t) = \sum_u b_t(u) \mathcal{R}(s_t, a_t \mid u). \tag{2}$$

The LLM agent aims to optimize the expected cumulative reward starting from an initial observable prompt $s_0$, and an initial belief $b_0$, represented by the value function:

$$V^\pi(s_0, b_0) = \mathbb{E}\left[\sum_{t=0}^{\infty} \gamma^t \mathcal{R}^b(s_t, b_t, a_t) \mid \pi, s_0, b_0\right], \tag{3}$$

where $\pi : \mathcal{S} \times \Delta(\mathcal{U}) \to \mathcal{A}$ is the policy, and $\gamma \in [0, 1)$ is the discount factor. The expectation is taken over $a_t \sim \pi(\cdot \mid s_t, b_t)$ and $\tilde{s}_{t+1} \sim \tilde{\mathcal{T}}(\cdot \mid \tilde{s}_t, a_t)$.

### 3.2 Introducing Intrinsic Reward to Multi-Turn RLHF

Conventional methods for training LLMs struggle to identify the optimal policy under this formulation. This difficulty arises primarily from two challenges. First, personalization may require obtaining information about the user over the course of a conversation by learning about their preferences or constraints in an enjoyable way, and then using that information to make personalized recommendations, or adapt to their style. Therefore, whether the LLM has successfully personalized the conversation to the user typically can only be evaluated at the end of the conversation, resulting in an extremely sparse reward signal. This sparsity hinders the model's ability to learn which early-stage actions can lead to higher future personalized rewards. Second, there exists a data imbalance among different user groups within large corpora. As a result, the model tends to learn policies that perform well on the majority group [2], achieving relatively higher rewards while falling into a local minimum. This discourages further exploration associated with other users.

#### 3.2.1 Intrinsic Reward via User Modeling

To address this issue, we propose to introduce Intrinsic Motivation (IM) to train a language model that can actively learn about the user type "*out of curiosity*", and then adapt to the preference of each user. This intrinsic reward is given by the agent's improvement in belief over the user type across the turns. The intuition for this idea is that training the model to acquire information about the user type $u$ will better enable it to optimize the personalized reward $\mathcal{R}(s, a \mid u)$. Since we are in a large-scale language domain, we do not use traditional Bayes-based belief updates. Instead, we can leverage a parameterized **user model** that predicts the probability distribution over user types based on the conversation rollout. This user model can be either trained or prompted depending on the task. Specifically, the user model takes in the current conversation rollout $s_{t+1}$ after applying $a_t$ and sampling the user response, and predicts the belief $b_{t+1}(u) := b(u \mid s_{t+1})$ (which is a probability distribution over all user types). With this user modeling, we can define intrinsic rewards such as the improvement in accuracy over ground truth $b_{t+1}(u^*) - b_t(u^*)$ between two turns, or the entropy reduction $H(b_t) - H(b_{t+1})$ of the probability distribution.

The CURIO framework is illustrated in Figure 2, with four different LLMs involved in training. In each episode, the current *policy model* (which we are training) engages in a multi-turn conversation with a fixed, simulated *environment model*, which is meant to simulate a human user. The *reward model* employed in traditional RLHF evaluates the entire conversation, generating an extrinsic reward $\mathcal{R}^{\text{ext}}$ provided only at the end of the conversation. In contrast, the *user model* predicts the probability

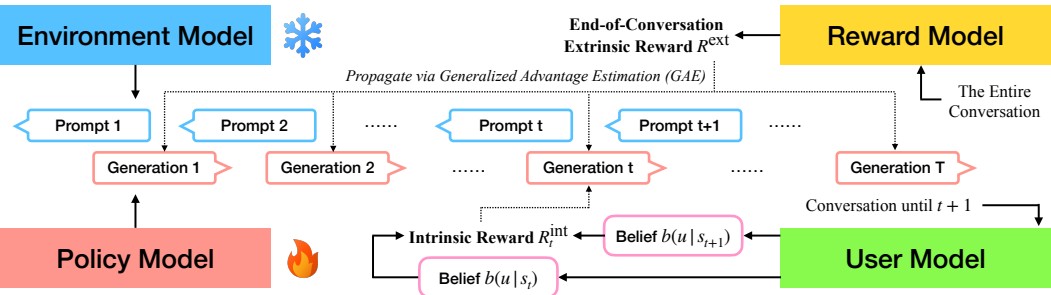

Figure 2: RL fine-tuning Pipeline for CURIO framework in one episode. We leverage a user model to obtain dense turn-based intrinsic rewards as a supplement to the sparse end-of-conversation rewards.

distribution over user types at each conversational turn, based on the dialogue context up to that point (i.e. $b_{t+1}(u)$). These probability distributions are then transformed into turn-based rewards $\mathcal{R}_t^{\text{int}}$. Consequently, this method supplements the original sparse reward structure with dense intrinsic rewards, effectively guiding the policy to better understand and adapt to various user types. For more details on how the models are trained with multi-turn RL, please refer to Appendix A.1.

### 3.2.2 Engineering Details

From an engineering perspective, both the environment and reward models are loaded directly into memory alongside the policy model, as all three operate at comparable scales. At the beginning of each episode, the policy and environment models generate conversation turns in an interleaved fashion. After the full conversation is generated, the reward model evaluates it, providing an extrinsic, end-of-conversation reward. This single reward is propagated to each turn as a value calculated from Generalized Advantage Estimation (GAE) [28]. To maintain efficiency and avoid excessive memory usage, the more computationally demanding user model is deployed separately and accessed via remote API calls. This approach enables batching predictions from different conversations and turns (time steps), allowing parallel computation of per-turn rewards. Subsequently, these per-turn intrinsic rewards are combined with previously calculated value, yielding a unified reward signal for each turn.

### 3.3 Relationship with Potential-based Reward Shaping

Potential-based Reward Shaping (PBRS) [16] is widely applied in traditional RL. It has been extensively studied within MDPs and later extended to the POMDP setting [17]. This series of work provides insights into how intrinsic rewards can be effectively designed. In particular, the following theorem offers fundamental justification for employing intrinsic rewards of specific forms.

**Theorem 1.** [17] Let $\phi : \Delta(\mathcal{U}) \to \mathbb{R}$ be a function defined over the belief distribution $b_t$. If we shape the agent's reward by adding the difference in the agent's belief between two subsequent timesteps

$$r^b(s_t, b_t, a_t) = \mathcal{R}^b(s_t, b_t, a_t) + \gamma\phi(b_{t+1}) - \phi(b_t), \tag{4}$$

where $\gamma$ is the discount factor, then optimizing $r^b$ yields the same policy as optimizing the original reward $\mathcal{R}^b$ in Eq. 3. In other words, **adding PBRS does not affect the optimal policy.**

Intuitively, with a better user prediction, the policy can better tailor its actions to achieve higher returns. In our formulation, we treat one conversation utterance as a timestep, so we can reward the agent for improving its belief about the user between two subsequent conversation turns, after it has incorporated their response to its last question or statement. We propose to use the following functions that incentivize the improvements in user prediction:

$$\phi_{\text{acc}}(b) = b(u^*), \quad \phi_{\text{log-acc}}(b) = \log b(u^*), \quad \phi_{\text{neg-ent}}(b) = -H(b) = \sum_u b(u) \log b(u), \tag{5}$$

corresponding to accuracy, logarithm of accuracy, and negative entropy, respectively. Noting that adding an auxiliary reward that follows the formulation above does not change the optimal behavior of the policy according to the theorem, we hypothesize that it just potentially make the policy *easier to learn*, since it directly encourages accurate inference of user types. We further conduct a case study on a simplified setting to theoretically demonstrate the effectiveness of such rewards in Appendix C.

In this paper, we experiment with the following intrinsic rewards that incentivize learning about the user. The first column shows a set of PBRS rewards, which are in a differential (**Diff**) format corresponding to Equation 5. The second column shows a second set of possible intrinsic user curiosity rewards that are PBRS terms, but nevertheless may be reasonable objectives. For example, Information Gain is the mutual information between the random variable $S_{t+1}$ and $u$, which can be written as the KL divergence $D_{\mathrm{KL}}[b_{t+1}(u)||b_t(u)]$ practically (after sampling $s_{t+1}$) according to Houthooft et al. [13]. The second column cannot guarantee the optimality of policy learning.

| | Potential-based Reward Shaping | | Other Reward Shaping |
|---|---|---|---|
| **DiffAcc** | $\gamma b_{t+1}(u^*) - b_t(u^*)$ | **Acc** | $b_{t+1}(u^*) - 1/|\mathcal{U}|$ |
| **DiffLogAcc** | $\gamma \log b_{t+1}(u^*) - \log b_t(u^*)$ | **Ent** | $\log |\mathcal{U}| - H(b_{t+1})$ |
| **DiffEnt** | $H(b_t) - \gamma H(b_{t+1})$ | **InfoGain** | $D_{\mathrm{KL}}[b_{t+1}(u)||b_t(u)]$ |

## 4   Experiments

To comprehensively evaluate our method's ability to personalize conversations across diverse scenarios, we conducted experiments using two distinct domains, which are each intended to answer specific research questions. To answer, *Can CURIO improve performance on personalization tasks?*, we designed a conversational personalization task Exercise Recommendation. In this task, the agent recommends an appropriate exercise strategy tailored to the user's lifestyle, health condition, and other attributes. Next, to investigate *Can CURIO effectively personalize conversations when personalization is not the ultimate objective?*, and *How does user learning affect conversation quality?*, we applied the method to an existing task Education Dialogue.

### 4.1   Exercise Recommendation

We first consider a case where **personalization is the main objective** of the conversation. Our core research question is whether multi-turn RL with improved user modeling as a turn-based intrinsic reward can enhance LLMs' ability to learn about the user, thereby improving personalization performance beyond that achieved by training solely on a sparse, final end-of-conversation reward.

To study this, we design a new task, Exercise Recommendation, where the agent provides personalized recommendations, similar to conversational recommender applications. In this scenario, the agent functions as a health advisor, tasked with recommending personalized fitness strategies tailored to each user. To enhance realism, we designed a comprehensive list of user attributes [29] encompassing multiple aspects such as lifestyle, socioeconomic status, and health conditions etc. Consequently, to make personalized recommendations, the agent must elicit user information and preferences through multiple rounds of dialogue before choosing a strategy at the end of the conversation.

Each user has a particular backstory and a ground truth label of which exercise strategy would be most effective for them, based on their profile. For both training and inference, the agent is only rewarded when its recommendation is aligned with the user's ground-truth strategy after the whole conversation. We assume the availability of a relatively accurate user model capable of inferring user type from the current conversation. Such models can be developed in real applications by training a user classifier on user behaviors and final choices (e.g., clicks). The dataset construction involved three steps: (1) *User Attribute Definition and Sampling:* For each user, we randomly sample values for 20 attributes encompassing various personal characteristics. (2) *Ideal Strategy Derivation:* We define a set of 8 exercise strategies and establish a deterministic logic rule that maps user attributes to an ideal (ground-truth) strategy (see Appendix D.2). For example, we may recommend a team sport for those who are outdoorsy and extroverted. (3) *User Backstory Generation:* We utilize the Gemini model to generate a detailed backstory for each user based on their attribute values. Each simulated user is prompted only with the backstory. Please refer to Appendix D for more details.

### 4.2   Education Dialogue

In many other tasks, however, personalization can improve performance on the task but **is only one component of the task rather than the only aim**. These tasks are helped by accurate user modeling but usually have a more complicated reward function. For example, in teaching scenarios, knowing a student's learning style or knowledge level is critical to helping the student, but the agent must

| Baseline | | Other Reward Shaping | | | Potential-based Reward Shaping | | |
|---|---|---|---|---|---|---|---|
| **SFT** | **MTRLHF** | **InfoGain** | **Ent** | **Acc** | **DiffEnt** | **DiffLogAcc** | **DiffAcc** |
| 54.0 | 68.5(+14.5) | 63.0(+9.0) | 82.0(+28.0) | 84.0(+30.0) | 84.0(+30.0) | 86.0(+32.0) | **87.5(+33.5)** |

Table 1: Success Rates (%) of different models over Exercise Recommendation. The models are evaluated by whether the predicted exercise strategy is the same as the pre-defined target. The values in brackets are the improvement over SFT. Overall, CURIO significantly improves the success rates.

still be an effective teacher and explain concepts clearly. Furthermore, the *reward model is not user-conditioned* at all in many real-world applications, so we hypothesize that simply introducing the intrinsic motivation can lead to a personalized dialogue agent in these scenarios as well.

We use the Education Dialog dataset introduced by Shani et al. [18], which simulates an educational setting where an LLM agent teaches students a given topic. This dataset is particularly valuable as it incorporates individual student learning preferences. We specifically selected two representative and contrasting learning styles: *story telling* and *hands-on activities*. These styles serve as distinct user preferences, allowing us to assess the agent's ability to adapt its conversational strategy.

Because Shani et al. [18] only evaluate the standard conversation quality, we establish a protocol to further evaluate personalization given that the student in each episode has a ground-truth preferred learning style. Specifically, all the models are evaluated across: (1) *personalization*, assessing the agent's ability to tailor conversations to user's ground-truth preference, and (2) *conversation quality*, determining whether personalization was achieved without compromising coherence and overall quality. Automated evaluation was performed using Gemini [30] to compare a pair of conversations generated by two models and choose the better response, and we use win rate as evaluation metrics. For conversation quality, we use the same prompt proposed by Shani et al. [18]. We further conduct a human evaluation study of both personalization and conversation quality on the Education Dialogue task and compare the results to our Auto Eval, in order to demonstrate the realism of the LLM-as-a-judge evaluation. Please refer to Appendix B.1 for more details.

### 4.3 Baselines and Model Usages

Our personalized conversation tasks are set in a multi-turn framework, and we build upon the multi-turn RLHF pipeline introduced by Shani et al. [18]. To the best of our knowledge, few existing works have explored multi-turn RL fine-tuning for personalization. We thus compare our method to a vanilla Multi-Turn RLHF (MTRLHF) baseline following their work. For RL fine-tuning, we use several LLM components: (1) the environment model and the initial policy model are SFT checkpoints of the Gemma 2B model [31]. For Exercise, we prompt Gemini to generate SFT data, while for Education, we directly use the checkpoints from the original work. (2) The value model used in RL fine-tuning is also initialized from Gemma 2B. (3) We use a scripted reward function for Exercise, comparing final-turn model outputs with ground truth targets, and adopt the reward model from Shani et al. [18] for Education. (4) We use prompted Gemma 7B as the user model. In Exercise, it infers user traits relevant to strategy selection and compute a probability distribution over strategies; in Education, it directly predicts the student's preferred learning style from the ongoing conversation.

## 5 Results

Overall, our results show that the CURIO method can significantly help with personalization on different tasks while maintaining the conversation quality. See Appendix F for example conversations.

**CURIO enhances personalization and reduces generalization gap.** Table 1 presents the success rates of different models over the Exercise Recommendation task, which is computed by sampling 200 conversations with the trained checkpoints. The initial SFT model achieved success rate of 54%. With furthur RL training on the task reward (recommendation success rate), traditional MTRLHF increases the rate to 68.5%. With CURIO, the success rate reached up to 87.5%, doubling the improvements of MTRLHF. For all the models, we picked the checkpoints with the highest validation performance within a total training budget of 30000 steps to prevent overfitting.

| | Baseline | Other Reward Shaping | | | Potential-based Reward Shaping | | |
|---|---|---|---|---|---|---|---|
| | **MTRLHF** | **InfoGain** | **Ent** | **Acc** | **DiffEnt** | **DiffAcc** | **DiffLogAcc** |
| **MTRLHF** | - | 93.04 | 55.70 | 7.91 | 51.90 | 42.72 | 24.05 |
| **InfoGain** | 6.96 | - | 42.41 | 0.00 | 29.11 | 9.18 | 0.63 |
| **Ent** | 50.00 | 57.59 | - | 39.56 | 43.35 | 49.05 | 44.62 |
| **Acc** | **92.09** | 100.00 | 60.44 | - | 70.57 | 85.13 | 64.87 |
| **DiffEnt** | 48.10 | 70.89 | 55.06 | 29.43 | - | 40.51 | 34.49 |
| **DiffAcc** | **57.28** | 90.82 | 50.95 | 14.87 | 59.49 | - | 34.81 |
| **DiffLogAcc** | **75.95** | 99.37 | 55.38 | 35.13 | 65.51 | 65.19 | - |

Table 2: Side-by-side Auto Eval Results on Personalization. Each entry is the win rate (%) of conversations generated with row method, over ones generated with column method. Our models with *accuracy-based* rewards outperform the baseline model in conducting personalized conversations.

During training, we observed that traditional methods are significantly impacted by a generalization gap. Figure 3 presents the training and evaluation accuracy curves, where the evaluation is performed on novel users simulated from backstories that were separately held out. The baseline model exhibits a pronounced trend of overfitting throughout the training. In contrast, our CURIO model demonstrates synchronized improvement in both curves before 15k steps, and significantly outperforms the baseline. We hypothesize that this is because the baseline model personalizes by memorizing mappings from superficial user details to specific strategies seen during training. Our models generalize more effectively to novel users because they are *learning how to learn* about the user during the conversation—asking informative questions that help distinguish between different user types.

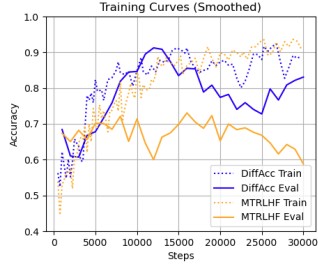

Figure 3: Training curves for Exercise Recommendation.

**CURIO remains effective when personalization is relevant but not ultimate goal.** Table 2 shows the pairwise win rates (judged by Gemini) across all the models on *personalization* over Education Dialogue. We can observe that, all the *accuracy-based* intrinsic rewards **significantly improve personalization** ability within the conversations. Notably, in our human study, an exact two-sided binomial test showed that humans chose DiffLogAcc over MTRLHF 75.75% of the time (p < .001), consistent with the Auto Eval rate of 75.95%, which supports the validity of Auto Eval.

To better demonstrate the different behavior in actively learning about user type, we show the oracle prediction accuracy of user type given the conversation stops at the third turn. This accuracy indicates whether the conversation generated by current policy and the simulated user can explicitly reveal information about the user. Figure 4 shows that our Differential Accuracy Model can learn to ask about the user type in the first few turns starting from 10k steps, and maintaining a prediction accuracy over 90%, while the baseline conversations only exhibit the ground truth user type around 70% of the time. Notably, the accuracy is higher than random guessing on baseline conversations mainly because the student often spells out their preferences directly without being asked. In short, the baseline rarely prompts users to disclose preferences, whereas our model actively explores to uncover user attributes. Furthermore, it serves as evidence that CURIO functions well without a perfect user model.

**CURIO with a proper reward choice preserves conversation quality.** Table 3 shows the pairwise win rates on *conversation quality*. Generally speaking, the CURIO models with potential-based reward shaping have a relatively smaller negative impact on the conversation quality. Among them, the Differential Log Accuracy reward is rated as significantly higher quality than baseline and all other intrinsic rewards. Note that the baseline is trained with the extrinsic reward, which is built from the preference pairs annotated by exactly the same prompt as Auto Eval process. This explains the baseline's inherent advantage in conversation quality. However, since the extrinsic reward is not user conditioned, optimizing it will actually hurt personalization.

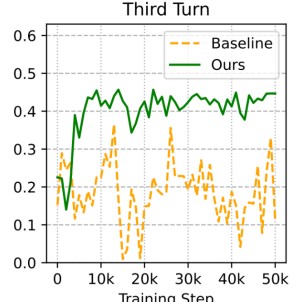

Figure 4: Calibrated user modeling accuracy for baseline vs CURIO model (DiffAcc) at the third turn in Education Dialogue. y-axis: $b(u^*) - 1/2$. The conversations generated with our model consistently reveal more user information.

| | Baseline | Other Reward Shaping | | | Potential-based Reward Shaping | | |
|---|---|---|---|---|---|---|---|
| | **MTRLHF** | **InfoGain** | **Ent** | **Acc** | **DiffEnt** | **DiffAcc** | **DiffLogAcc** |
| **MTRLHF** | - | 99.05 | 73.42 | 87.34 | 65.19 | 71.84 | 45.57 |
| **InfoGain** | 0.95 | - | 2.85 | 5.38 | 0.95 | 4.11 | 0.00 |
| **Ent** | 26.58 | 97.15 | - | 62.34 | 26.90 | 57.59 | 23.10 |
| **Acc** | 12.66 | 94.62 | 37.66 | - | 19.62 | 42.72 | 13.61 |
| **DiffEnt** | 34.81 | 99.05 | 73.10 | 80.06 | - | 73.73 | 31.65 |
| **DiffAcc** | 28.16 | 95.89 | 42.41 | 56.65 | 26.27 | - | 18.99 |
| **DiffLogAcc** | **54.43** | **100.00** | **76.90** | **86.39** | **68.35** | **81.01** | - |

Table 3: Side-by-side Auto Eval Results on Conversation Quality. Each entry is the win rate (%) of conversations generated with row method, over ones generated with column method. Our models with *Potential-based Reward Shaping* have a smaller negative impact on conversation quality, and **DiffLogAcc** outperforms the baseline.

**Addressing Reward Hacking.** In order to train multi-turn RL models at scale, we use LLMs to simulate the user and act as reward models. A limitation of this approach is that RL-based methods which attempt to maximize rewards can sometimes engage in "*reward hacking*", exploiting weaknesses in either the reward model or user model to obtain higher rewards in ways that do not correspond to desirable behaviors. With an extrinsic reward model that is not user-conditioned, the baseline multi-turn RL model adopts a *merging* teaching style called "role-playing video", which is not one of the true learning styles, but results in a spuriously high extrinsic reward. Similarly, when using the entropy-based intrinsic rewards that are not "grounded" (i.e., are only based on the classifier's certainty and do not make use of a ground-truth user label), we observe that the models perform really well on one particular user type, but badly on the other. For example, even though the student has expressed preference in story-telling, the teacher insists on hands-on style. We attribute it to the emergence of "controlling behavior", where the policy attempts to convince the classifier that the user belongs to one particular type, rather than actually adhering to the ground-truth type. The problem with InfoGain is more serious, where the policy maximizes KL divergence by inducing sharp shifts in the predicted user type distribution between consecutive turns. However, by using our proposed accuracy-based rewards, which require predicting the actual user type rather than tricking the user classifier, we can resolve these issues and attain better performance.

Another form of reward hacking happens with non-potential-based rewards, such as Acc and Ent, where the policy model learns to arbitrarily increase the length of the conversation because the intrinsic reward is assigned to each turn on the same scale, thus hurting the conversation quality. This emphasizes the necessity of using potential-based intrinsic rewards.

## 6 Conclusions

This paper introduced a novel framework CURIO for enhancing personalization in LLMs on multi-turn conversation tasks. By leveraging user modeling and integrating intrinsic rewards into multi-turn reinforcement learning, our approach encouraged the LLM to actively learn user traits and adapt its responses accordingly. Experiments across two distinct domains demonstrate that CURIO improves personalization in multi-turn conversations across various scenarios, whether personalization is the ultimate or a partial goal, while maintaining conversation quality.

**Limitations.** Our framework assumes pre-defined and static user traits, which may not reflect the complexities of real-world conversations. Furthermore, our experiments currently rely on LLM-based user simulators for both training and testing. This approach was necessitated by the scarcity of large-scale, open-source datasets for conversational personalization and the impracticality of using live human interaction for training. A primary consequence of this simulation is the potential discrepancy between synthetic reward signals and authentic human preferences.

**Societal Impact.** Using ungrounded personalization with CURIO method may lead to negative impacts such as controlling behavior, but grounded personalization is generally safer in many real-world tasks. Future research should explore personalization within more complex, temporally-evolving contexts, and build conversational agents that can achieve robust zero-shot pluralistic alignment while ensuring ethical considerations like privacy, transparency, and bias mitigation.

**Acknowledgement.** We would like to express our sincere gratitude to all those who contributed to the completion of this work. We particularly wish to thank Howard Zhou, Mojtaba Seyedhosseini, Andrew Lingao, Joel Z Leibo, Shawn O'Banion, Jun Xie, Jianxun Lian, Yulia Tsvetkov, and Sergey Levine for providing the necessary resources, invaluable guidance and support throughout the entire process. We would like to thank Mickel Liu, Yancheng Liang, Weihang Xu, Arnab Maiti, Kunal Jha, Sriyash Poddar, Carrie Yuan, Shakti Senthil, Shanfeng Zhang, Lin Ning, Luyang Liu, Cecilia Tilli for insightful discussions and constructive feedback that helped improve the clarity and depth of this work. This research was supported by the Cooperative AI Foundation, the UW-Amazon Science Gift Hub, Sony Research Award, UW-Tsukuba Amazon NVIDIA Cross Pacific AI Initiative (XPAI), the Microsoft Accelerate Foundation Models Research Program, Character.AI, DoorDash, and the Schmidt AI2050 Fellows program. This material is based upon work supported by the Defense Advanced Research Projects Agency and the Air Force Research Laboratory, contract number(s): FA8650-23-C-7316. Any opinions, findings and conclusions, or recommendations expressed in this material are those of the author(s) and do not necessarily reflect the views of AFRL or DARPA.

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

# A  Training Details

## A.1  Optimization of Rewards with Multi-Turn RL

CURIO method introduces two different reward signals: extrinsic reward $\mathcal{R}^{\text{ext}}$ and intrinsic reward $\mathcal{R}^{\text{int}}_t$. To ensure consistency, we denote $\mathcal{R}^{\text{ext}}_t$ as the extrinsic reward obtained at each turn, where $\mathcal{R}^{\text{ext}}_t = 0$ for $t < T$. We further introduce a KL divergence penalty term $D_{\text{KL}}(\pi_\theta \| \pi_{\text{ref}})$ against the fixed reference model, following Jaques et al. [32]. We use multiple hyperparameters to balance different reward signals, and the total reward is

$$r_t = \alpha_{\text{ext}} \mathcal{R}^{\text{ext}}_t + \alpha_{\text{int}} \mathcal{R}^{\text{int}}_t - \beta D_{\text{KL}}(\pi_\theta \| \pi_{\text{ref}}) \tag{6}$$

Following Shani et al. [18], we apply Generalized Advantage Estimation (GAE) [28] to our total rewards, which aims to propagate end-of-conversation extrinsic reward $\mathcal{R}^{\text{ext}}$ to turn-level signals. In particular, we need a stand-alone value model to estimate the value function $V(s_t)$ for each turn. The propagated reward signals for each turn is then given by

$$\hat{r}_t = \sum_{t'=t}^{T} (\gamma\lambda)^{t'-t} [r_{t'} + \gamma(1-\lambda)V(s_{t'+1})], \tag{7}$$

where $\gamma$ is the discount factor, $\lambda$ is the coefficient for GAE, and define $V(s_{T+1}) = 0$. We then use a REINFORCE [33] policy gradient training approach to optimize the policy model $\pi_\theta$ over these propagated rewards $\hat{r}_t$. Formally, the policy is optimizing the objective: $\max_\theta \mathbb{E}_{\pi_\theta}[\hat{r}_t]$.

## A.2  Model Choices

The personalized conversation tasks are in a multi-turn setting, we leverage the multi-turn RLHF pipeline implemented by Shani et al. [18]. In our RL fine-tuning process, we use the following models:

- **Environment Model and (Initial) Policy Model**: Since we are training the multi-turn policy, We leverage SFT LLM checkpoints (Gemma 2B model [31]) to serve as the initial policy checkpoint and simulate the users. For Exercise task, we prompted Gemini 1.5 Pro [30] to generate conversational data for supervised fine-tuning. For Education task, we directly use the checkpoints in the original work.

- **Value Model**: The Value model for value estimation is initialized by Gemma 2B.

- **Reward Model/Function**: For Exercise task, we used a scripted reward function that directly compares the model generations in the final turn with the ground truth targets, no reward model. For Education task, we directly adopt the reward model developed by [18].

- **User Model**: For Exercise task, we employ a Gemma 7B model [31] to predict the answers to a series of user traits that are relevant to the optimal strategy from the conversation so far, and then compute the probability distribution over all strategies. For Education task, we use the Gemma 7B model to directly predict the student's preferred learning style based on the ongoing conversation.

## A.3  Hyperparameters

We followed the training recipe and hyperparameters from Shani et al. [18]. On top of the original extrinsic reward, we added intrinsic reward to each turn of the conversation as described above, with a coefficient coefficient weight $\alpha_{\text{int}}$ on intrinsic reward when adding to the extrinsic reward to balance the scale of extrinsic and intrinsic rewards. For Education Dialogue, we choose $\alpha_{\text{int}} = 9.0$ for all the settings in Education Dialogue, with other hyperparameters listed in Table 4. For all the settings, we select several checkpoints that has the highest intrinsic rewards before 30k steps, and then choose the one that performs the best on conversation quality. For Exercise Recommendation, we choose $\alpha_{\text{int}} = 5.0$ for **DiffAcc** and **DiffEnt**, $\alpha_{\text{int}} = 1.0$ for **Acc**, **Ent**, and **DiffLogAcc**, and $\alpha_{\text{int}} = 0.1$ for **InfoGain**. The other hyperparameters are listed in Table 5. The user classifier temperature $tau$ is used when calculating the probability distribution over user types from logits using Softmax function. For all the settings including baselines, we select the checkpoints that has the highest validation extrinsic reward (strategy prediction accuracy) before 30k steps to avoid overfitting. Note that the number of turns for Exercise Recommendation includes several conversational turns and a final turn for strategy prediction.

| Hyperparameters for Education Dialogue | |
|---|---|
| Policy Model Learning Rate $\eta_{\text{policy}}$ | 4e-7 |
| Value Model Learning Rate $\eta_{\text{valuey}}$ | 4e-7 |
| Batch Size $B$ | 16 |
| KL Regularization Coefficient $\beta$ | 0.01 |
| GAE Coefficient $\lambda$ | 0.95 |
| Turn Discount $\gamma$ | 0.95 |
| Max Number of Turns $T$ | 10 |
| Extrinsic Reward Weight $\alpha_{\text{ext}}$ | 1.0 |
| User Classifier Temperature $\tau$ | 5.0 |

Table 4: Hyperparameters for Education Dialogue.

| Hyperparameters for Education Dialogue | |
|---|---|
| Policy Model Learning Rate $\eta_{\text{policy}}$ | 4e-7 |
| Value Model Learning Rate $\eta_{\text{valuey}}$ | 4e-7 |
| Batch Size $B$ | 16 |
| KL Regularization Coefficient $\beta$ | 0.02 |
| GAE Coefficient $\lambda$ | 0.95 |
| Turn Discount $\gamma$ | 0.95 |
| Max Number of Turns $T$ | 6 |
| Extrinsic Reward Weight $\alpha_{\text{ext}}$ | 3.0 |
| User Classifier Temperature $\tau$ | 5.0 |

Table 5: Hyperparameters for Exercise Recommendation.

# B  Extended Results

## B.1  Human Study for Education Dialogue

We conducted a human evaluation study of both personalization and conversation quality on the Education Dialogue task. We invited 10 students to participate. Each participant was asked to evaluate dialogues generated by three models across 20 different settings (topic/user type). Evaluations were conducted through pairwise comparisons across two dimensions: personalization and conversation quality, resulting in a total of 200 pairwise comparisons per dimension.

An exact two-sided binomial test showed that human evaluators chose DiffLogAcc over the MTRLHF baseline in 152 / 200 pairwise comparisons on personalization (preference = 0.76, 95% CI [0.69, 0.82]), p < .001. Importantly, increase in personalization did not reduce conversation quality: DiffLogAcc was preferred in 90 / 200 quality judgements (preference = 0.45, 95% CI [0.38, 0.52]), and the same exact test found no significant difference from chance (p = .179). In Table 6, we provide two tables containing the full results of the human study, which correspond to Table 2 and 3. We note that the human evaluation results are reasonably consistent with those obtained through Auto Eval.

## B.2  Results for Education Dialogue across Different User Types

We present the Auto Eval results for Education Dialogue on two different user types in Table 7 and Table 8. We found that the pairwise win rates of these models over personalization can vary significantly when the user ground truth label is different. The values in red color are the win rates of entropy-based models against the baseline, where both models are performing very well on the second user type, but extremely badly on the first one. This further supports our observation that agents trained with entropy-based reward shaping functions may converge to one particular user type that is not necessarily correct. That is the reason of using accuracy-based reward shaping functions. For conversation quality, this phenomenon is no longer significant, and the **DiffLogAcc** model is still outperforming all the other models.

| Personalization | MTRLHF | DiffAcc | DiffLogAcc |
|---|---|---|---|
| **Baseline** | - | 30.50 | 24.25 |
| **DiffAcc** | 69.50 | - | 46.75 |
| **DiffLogAcc** | 75.75 | 53.25 | - |

| Conv. Quality | MTRLHF | DiffAcc | DiffLogAcc |
|---|---|---|---|
| **Baseline** | - | 60.25 | 55.00 |
| **DiffAcc** | 39.75 | - | 46.25 |
| **DiffLogAcc** | 45.00 | 53.75 | - |

Table 6: Human study results on Education Dialogue.

### B.3 Questions Distribution for Exercise Recommendation

In the main paper, we hypothesize that the severe overfitting of the baseline model is because the baseline model personalizes by memorizing mappings from superficial user details to specific strategies seen during training. Our models generalize more effectively to novel users because they are *learning how to learn* about the user during the conversation—asking informative questions that help distinguish between different user types. In Figure 5 we present the questions distribution of CURIO model, RLHF baseline, and SFT initial checkpoint. Note that Occupation is a relevant attribute because it gives us an idea about the Socioeconomic Status. The SFT and RLHF model are asking about and memorizing irrelevant attributes like name and hobbies, while our CURIO model is able to find a key attribute–introverted vs extroverted–during the training process. None of the models is able to ask about motivation, probably because this attribute is only helpful in distinguishing between Strategy 7 and 8. See section D for full details of user traits and strategies.

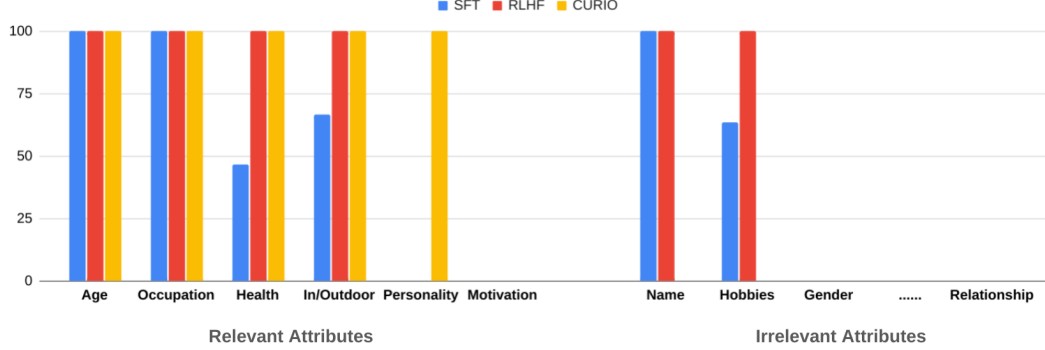

Figure 5: Questions distribution of CURIO model, RLHF baseline, and SFT initial checkpoint. The left ones are relevant attributes, and the right ones are irrelevant to the strategy recommendation.

## C Case Study: Multi-Turn Conversation as Combinatorial Bandits

In this section, we briefly discuss how the personalized multi-turn conversation problem can be connected to some existing theoretical frameworks. In particular, we will simplify the Exercise Recommendation task into a combinatorial bandits problem, and provide an insight on why CURIO method can help improve the efficiency.

### C.1 High-level Intuition

A multi–turn conversation of fixed length $H$ is viewed as a single *episode*. During the episode the agent asks $H$ questions. We assume the order of these actions does not affect the final utility, since the reward is only given at the end of the conversation based on the final recommendation. Therefore,

| First User Type | Baseline | Other Reward Shaping | | | Potential-based Reward Shaping | | |
|---|---|---|---|---|---|---|---|
| | MTRLHF | InfoGain | Ent | Acc | DiffEnt | DiffAcc | DiffLogAcc |
| **MTRLHF** | - | 0.96 | 1.00 | 0.08 | 1.00 | 0.37 | 0.27 |
| **InfoGain** | 0.04 | - | 0.85 | 0.00 | 0.58 | 0.00 | 0.00 |
| **Ent** | 0.00 | 0.15 | - | 0.00 | 0.06 | 0.00 | 0.00 |
| **Acc** | **0.92** | 1.00 | 1.00 | - | 0.99 | 0.85 | 0.66 |
| **DiffEnt** | 0.00 | 0.42 | 0.91 | 0.01 | - | 0.00 | 0.01 |
| **DiffAcc** | **0.63** | 1.00 | 1.00 | 0.15 | 1.00 | - | 0.43 |
| **DiffLogAcc** | **0.73** | 1.00 | 1.00 | 0.34 | 0.99 | 0.57 | - |

| Second User Type | Baseline | Other Reward Shaping | | | Potential-based Reward Shaping | | |
|---|---|---|---|---|---|---|---|
| | MTRLHF | InfoGain | Ent | Acc | DiffEnt | DiffAcc | DiffLogAcc |
| **MTRLHF** | - | 0.90 | 0.11 | 0.08 | 0.04 | 0.49 | 0.22 |
| **InfoGain** | 0.10 | - | 0.00 | 0.00 | 0.00 | 0.18 | 0.01 |
| **Ent** | **1.00** | 1.00 | - | 0.79 | 0.81 | 0.98 | 0.89 |
| **Acc** | **0.92** | 1.00 | 0.21 | - | 0.42 | 0.85 | 0.64 |
| **DiffEnt** | **0.96** | 1.00 | 0.19 | 0.58 | - | 0.81 | 0.68 |
| **DiffAcc** | **0.51** | 0.82 | 0.02 | 0.15 | 0.19 | - | 0.27 |
| **DiffLogAcc** | **0.78** | 0.99 | 0.11 | 0.36 | 0.32 | 0.73 | - |

Table 7: Personalization Auto Eval results for Education Dialogue on two different user types. For each row's model, the values represent the percentage of wins it achieved against the model specified in each column.

the whole episode can be abstracted as the selection of an unordered subset of size $k := H$ from a finite action catalogue. The delayed scalar outcome (customer satisfaction, purchase, etc.) arrives *after* the entire dialogue and cannot be decomposed into per-turn signals observable by the agent. Hence only the aggregate reward $R_t$ is available, matching the full-bandit feedback assumption.

### C.2  Theoretical Formulation

- **Base arms.** Let $\mathcal{A} = \{1, \ldots, K\}$ be the catalogue, $K \gg 1$. Each arm $a \in \mathcal{A}$ corresponds to a question or any atomic conversational move.

- **Episode action.** In episode $t = 1, 2, \ldots$, the agent selects a *super-arm* $S_t \subseteq \mathcal{A}$ with fixed cardinality $|S_t| = k$. The choice $S_t$ encodes the entire conversation, since we assume that the response given by the environment is deterministic.

- **Latent structure.** There exists an unknown subset of *useful* arms $U^\star \subseteq \mathcal{A}$ of size $m \ll K$.

- **Reward signal.** After pulling $S_t$ the agent receives a scalar reward $R_t$. The reward can be defined in various ways depending of the problem structure. Here are two simplified settings.

  1. **Additive (linear) reward**

  $$R_t = \sum_{a \in S_t} X_{a,t}, \qquad \mathbb{E}[X_{a,t}] = \mu_a \in \{0, 1\}.$$

  Only useful arms have $\mu_a = 1$; non-useful arms have $\mu_a = 0$. The $X_{a,t}$ can be i.i.d Gaussian random variables with the given mean and some fixed noise.

  2. **All-or-nothing (conjunctive) reward**

  $$R_t = \mathbf{1}\{U^\star \subseteq S_t\},$$

  i.e. $R_t = 1$ iff every useful arm is contained in the chosen super-arm (equivalently $S_t = U^\star$ when $m = k$), and 0 otherwise. Note that we assume $m \leq k$ in this case.

- **Feedback type.** The agent only observes $R_t$; it *does not* observe the individual $X_{a,t}$ or which arms were responsible for the reward. This is the *full-bandit* feedback setting.

| First User Type | Baseline | Other Reward Shaping | | | Potential-based Reward Shaping | | |
|---|---|---|---|---|---|---|---|
| | **MTRLHF** | **InfoGain** | **Ent** | **Acc** | **DiffEnt** | **DiffAcc** | **DiffLogAcc** |
| **MTRLHF** | - | 0.99 | 0.73 | 0.88 | 0.65 | 0.78 | 0.51 |
| **InfoGain** | 0.01 | - | 0.02 | 0.04 | 0.01 | 0.01 | 0.00 |
| **Ent** | 0.27 | 0.98 | - | 0.62 | 0.30 | 0.67 | 0.39 |
| **Acc** | 0.12 | 0.96 | 0.38 | - | 0.18 | 0.45 | 0.22 |
| **DiffEnt** | 0.35 | 0.99 | 0.70 | 0.81 | - | 0.75 | 0.42 |
| **DiffAcc** | 0.22 | 0.99 | 0.33 | 0.54 | 0.25 | - | 0.20 |
| **DiffLogAcc** | 0.49 | **1.00** | **0.61** | **0.78** | **0.58** | **0.80** | - |

| First User Type | Baseline | Other Reward Shaping | | | Potential-based Reward Shaping | | |
|---|---|---|---|---|---|---|---|
| | **MTRLHF** | **InfoGain** | **Ent** | **Acc** | **DiffEnt** | **DiffAcc** | **DiffLogAcc** |
| **MTRLHF** | - | 0.99 | 0.74 | 0.87 | 0.66 | 0.66 | 0.41 |
| **InfoGain** | 0.01 | - | 0.04 | 0.07 | 0.01 | 0.07 | 0.00 |
| **Ent** | 0.26 | 0.96 | - | 0.63 | 0.23 | 0.48 | 0.08 |
| **Acc** | 0.13 | 0.93 | 0.37 | - | 0.21 | 0.41 | 0.06 |
| **DiffEnt** | 0.34 | 0.99 | 0.77 | 0.79 | - | 0.73 | 0.22 |
| **DiffAcc** | 0.34 | 0.93 | 0.52 | 0.59 | 0.27 | - | 0.18 |
| **DiffLogAcc** | **0.59** | **1.00** | **0.92** | **0.94** | **0.78** | **0.82** | - |

Table 8: Conversation Quality Auto Eval results for Education Dialogue on two different user types. For each row's model, the values represent the percentage of wins it achieved against the model specified in each column.

- **Learning objective.** Denote by $\pi_t$ the selection rule for $S_t$. Since we are considering the LLM training process, the objective should be the best-subset identification in pure exploration setting. The policy needs to find $\widehat{U}$ such that $\Pr\{\widehat{U} = U^\star\} \geq 1 - \delta$ while minimising the sample complexity (number of episodes).

### C.3 Per–Turn Reward Shaping as Semi-Bandit Feedback

Assume we augment the delayed episode–level reward with an *intrinsic* reward signal delivered immediately after each conversational move:

$$r_t(a) = \mathbf{1}\{a \text{ is useful}\} \in \{0, 1\}, \qquad a \in S_t,$$

where $S_t$ is the size-$k$ set of actions chosen at turn $t$ of the dialogue. This shaping term informs the agent, right away, whether a selected action belongs to the useful action set $U^\star$. Our CURIO method is basically introducing this type of reward signals (where a reward is not exactly 1 but the gain in probability of the correct user type or the drop in entropy of the probability distribution), since the model can learn whether each of its action help improve the user prediction.

The shaped reward $\{r_t(a) : a \in S_t\}$ is *exactly* the definition of semi-bandit feedback: arm-level outcomes are revealed for the arms that were played, while arms not in $S_t$ remain unobserved. Hence the shaped conversational task and the canonical semi-bandit model share identical information structure and can be analysed similarly.

## D   Task Design for Exercise Recommendation

We generate 1000 simulated users, and split them into 800 for training and 200 for evaluation. Each user is mapped to one particular ground truth strategy among 8 different exercise strategies.

1. **User Attribute Definition and Sampling:** We defined 20 user attributes encompassing a range of personal characteristics. For each simulated user, we randomly sampled values for each of these attributes, creating a diverse user population.

2. **Ideal Strategy Derivation:** We established a deterministic logic rule that maps user attributes to an ideal (ground-truth) exercise strategy. For example, we may recommend a team sport for those who are outdoorsy and extroverted. The mapping rules are listed in the appendix. Among the 20 defined attributes, 5 were designated as relevant factors influencing the recommendation, while the remaining 15 served as background characteristics, emulating the complexity of real-world users.

3. **User Backstory Generation:** To provide contextual richness and ensure consistent agent behavior, we utilized the Gemini 1.5 Pro model [30] to generate a detailed backstory for each user based on their attribute values. These backstories were then used in prompts for the environment model, ensuring that the environment model remained consistent with the user's defined characteristics.

## D.1 List of User Attributes

- Name: from 1000 random names
- Age: randomly sampled between 15 and 65
- **Socioeconomic Status: randomly sampled from (low, medium, high)**
- Relationship Status: randomly generated
- Location From: randomly generated
- Occupation: randomly generated
- Education: randomly generated
- Religion: randomly generated
- Language Spoken: randomly generated
- **Have injuries or physical limitations: randomly sampled from (True, False)**[1]
- **Personality: randomly sampled from (introverted, extroverted)**
- **Motivation on plans: randomly sampled from (highly motivated, struggling with motivation)**
- **Enjoy outdoor or indoor activities: randomly sampled from (outdoorsy, indoorsy)**
- Hobbies and Interests: randomly generated
- Gender Identity: randomly generated
- Political Views: randomly generated
- Places Traveled: randomly generated
- Pet Ownership: randomly generated
- Sibling Information: randomly generated
- Life Goals and Ambitions: randomly generated

## D.2 Logic Rules for Optimal Exercise Strategy Recommendation

1. Recommend walking in parks: For those who have injuries and are outdoorsy.

2. Recommend yoga or tai chi at home: For those who have injuries and prefer staying indoors.

3. Recommend jogging or hiking: For those who do not have injuries, are outdoorsy, and are introverted.

4. Recommend a team sport: For those who do not have injuries, are outdoorsy, and are extroverted.

5. Offer a discount on a gym membership: For those who do not have injuries, prefer indoor activities, and have a low socioeconomic status.

6. Recommend home gym equipment: For those who do not have injuries, prefer indoor activities, have a higher socioeconomic status, are introverted, and are highly motivated.

---

[1]We manually set "Have injuries or physical limitations" to be True if the user's age is at least 55.

7. Recommend a personal trainer at the gym: For those who do not have injuries, prefer indoor activities, have a higher socioeconomic status, are introverted, and struggle with motivation.

8. Recommend a group class at the gym: For those who do not have injuries, prefer indoor activities, have a higher socioeconomic status, and are extroverted.

### D.3  Code for User Generation

```python
import numpy as np

def get_dict_str(input_str):
  input_str = input_str.strip()
  left = input_str.find('{')
  right = input_str.rfind('}')
  return input_str[left:right+1]

def generate_user_profile(name=None):
  """Generates a user profile.

  Args:
    name: The name of the user.
    use_gemma: Whether to use gemma to generate the profile.

  Returns:
    A tuple of the profile and a dictionary of the useful information.
  """
  socioeconomic_status = np.random.choice(
      ['low', 'medium', 'high'], p=[0.2, 0.6, 0.2]
  ).item()
  age = np.random.randint(15, 65)
  profile = (
      '{"name": "%s", "age": %d, "socioeconomic_status": "%s",'
      ' "relationship_status": <relationship>, "location_from": <location>,'
      ' "occupation": <occupation>, "education": <education>, "religion":'
      ' <religion>, "language spoken": <language spoken>}'
      % (name, age, socioeconomic_status)
  )
  demographic_prompt = (
      'Here is a profile for a random person in json format.\n'
      + profile
      + '\nPlease randomly generate the demographic information for them and'
      ' fill in blank in the json format. Output the json format only. '
  )
  profile = generate_by_LLM(demographic_prompt)
  profile = get_dict_str(profile)
  if age >= 55:
    have_injuries_or_physical_limitations = True
  else:
    have_injuries_or_physical_limitations = np.random.choice(
        [True, False], p=[0.1, 0.9]
    ).item()
  personality = np.random.choice(
      ['introverted', 'extroverted'], p=[0.6, 0.4]
  ).item()
  motivation_on_plans = np.random.choice(
      ['highly motivated', 'struggling with motivation'], p=[0.5, 0.5]
  ).item()
  enjoy_outdoor_or_indoor_activities = np.random.choice(
      ['outdoorsy', 'indoorsy'], p=[0.4, 0.6]
  ).item()
  profile = (
      profile[:-1]
      + ', "have_injuries_or_physical_limitations": %s, "personality": "%s",'
      ' "motivation_on_plans": %s, "enjoy_outdoor_or_indoor_activities": "%s"}'
      % (
```

```
                str(have_injuries_or_physical_limitations).lower(),
                personality,
                motivation_on_plans,
                enjoy_outdoor_or_indoor_activities,
            )
    )
    profile = (
        profile[:-1]
        + ', "hobbies_and_interests": <hobbies_and_interests>,'
        ' "gender_identity": <gender_identity>, "political_views":'
        ' <political_views>, "places_traveled": <places_traveled>,'
        ' "pet_ownership": <pet_ownership>, "sibling_information":'
        ' <sibling_information>, "life_goals_and_ambitions":'
        ' <life_goals_and_ambitions>}'
    )
    personal_info_prompt = (
        'Here is a profile for a random person in json format.\n'
        + profile
        + '\nPlease randomly generate the personal information for them and fill'
        ' in blank in the json format. Output the json format only. '
    )
    profile = generate_by_LLM(personal_info_prompt)
    profile = get_dict_str(profile)
    useful_profile_dict = {
        'name': name,
        'age': age,
        'socioeconomic_status': socioeconomic_status,
        'have_injuries_or_physical_limitations': (
            have_injuries_or_physical_limitations
        ),
        'personality': personality,
        'motivation_on_plans': motivation_on_plans,
        'enjoy_outdoor_or_indoor_activities': enjoy_outdoor_or_indoor_activities,
    }
    return profile, useful_profile_dict

def get_ground_truth_strategy(profile_dict):
    if profile_dict['have_injuries_or_physical_limitations']:
        if profile_dict['enjoy_outdoor_or_indoor_activities'] == 'outdoorsy':
            return 1
        else:
            return 2
    else:
        if profile_dict['enjoy_outdoor_or_indoor_activities'] == 'outdoorsy':
            if profile_dict['personality'] == 'introverted':
                return 3
            else:
                return 4
        else:
            if profile_dict['socioeconomic_status'] == 'low':
                return 5
            else:
                if profile_dict['personality'] == 'introverted':
                    if profile_dict['motivation_on_plans'] == 'highly motivated':
                        return 6
                    else:
                        return 7
                else:
                    if profile_dict['personality'] == 'extroverted':
                        return 8

def generate_backstory(profile):
    prompt = (
        'Here is a profile for a random person in json format.\n'
        + profile
```

```
        + ’\nPlease write a backstory in first person view based on the given’
        ’ profile. Please note that your story needs to cover all the’
        ’ information, but it does not have to follow the order provided.\n’
    )
    prompt += (
        ’Output format -- output your response in json, where you provide the’
        ’ following: \n’
    )
    prompt += ’{"backstory": <STRING_OF_YOUR_BACKSTORY>}’
    backstory = generate_by_LLM(prompt)
    backstory = get_dict_str(backstory)
    return backstory
```

## D.4    Oracle Classifier for Exercise Recommendation

Note that we are using an oracle classifier as the user model in Exercise Recommendation. An LLM is involved to answer some True/False questions to help determine whether a particular user attribute has been revealed during the conversation. The classifier finally outputs a probability distribution over all possible user types.

```
self.questions = {
    ’low_SES’: (
        ’Is the person having a low socioeconomic status (very concerned’
        ’ about expenses)? Answer True, False, or Unknown.’
    ),
    ’injury’: (
        ’Does the person has a special physical’
        ’ limitation? Answer True, False, or Unknown.’
    ),
    ’extroverted’: (
        ’Is the person an extroverted person? Answer True, False, or Unknown.’
    ),
    ’motivation’: (
        ’Is the person motivated to take on new plans? Answer True, False, or’
        ’ Unknown.’
    ),
    ’outdoor’: (
        ’Is the person an outdoor person? Answer True, False, or Unknown.’
    ),
}

def _async_generate_by_LLM(self, queries):
  [YOUR CODE GOES HERE]
  return responses

def _get_probs(self, conversations):
  keys = list(self.questions.keys())
  prompt = (
      ’The following is the conversation between a service agent and a’
      ’ customer:\n’
  )
  queries = []
  for conversation in conversations:
    for key in keys:
      queries.append((
          prompt
          + conversation
          + ’ Please answer the following question about the customer: ’
          + self.questions[key]
          + ’ Answer: ’
      ))

  def get_probs_from_answers(answers):
    probs = list()
```

```python
    if answers['low_SES'] == -1:
      answers['low_SES'] = 0.2
    if answers['injury'] == -1:
      answers['injury'] = 0.25
    if answers['extroverted'] == -1:
      answers['extroverted'] = 0.4
    if answers['motivation'] == -1:
      answers['motivation'] = 0.5
    if answers['outdoor'] == -1:
      answers['outdoor'] = 0.4

    probs.append(answers['injury'] * answers['outdoor'])
    probs.append(answers['injury'] * (1 - answers['outdoor']))
    probs.append(
        (1 - answers['injury'])
        * answers['outdoor']
        * (1 - answers['extroverted'])
    )
    probs.append(
        (1 - answers['injury']) * answers['outdoor'] * answers['extroverted']
    )
    probs.append(
        (1 - answers['injury'])
        * (1 - answers['outdoor'])
        * answers['low_SES']
    )
    probs.append(
        (1 - answers['injury'])
        * (1 - answers['outdoor'])
        * (1 - answers['low_SES'])
        * (1 - answers['extroverted'])
        * answers['motivation']
    )
    probs.append(
        (1 - answers['injury'])
        * (1 - answers['outdoor'])
        * (1 - answers['low_SES'])
        * (1 - answers['extroverted'])
        * (1 - answers['motivation'])
    )
    probs.append(
        (1 - answers['injury'])
        * (1 - answers['outdoor'])
        * (1 - answers['low_SES'])
        * answers['extroverted']
    )
    probs = np.array(probs)
    return probs

queries_with_responses = []
for query in queries:
  queries_with_responses.append([query, 'True'])
  queries_with_responses.append([query, 'False'])
  queries_with_responses.append([query, 'Unknown'])
responses = self._async_generate_by_LLM(
    queries_with_responses
)
prob_list = []
for conv_id in range(len(conversations)):
  answers = dict()
  for index, key in enumerate(keys):
    true_logits = responses[(conv_id * len(keys) + index) * 3]
    false_logits = responses[(conv_id * len(keys) + index) * 3 + 1]
    unknown_logits = responses[(conv_id * len(keys) + index) * 3 + 2]
    if true_logits > false_logits and true_logits > unknown_logits:
```

```
            answers[key] = 1
          elif false_logits > true_logits and false_logits > unknown_logits:
            answers[key] = 0
          else:
            answers[key] = -1
      probs = get_probs_from_answers(answers)
      prob_list.append(probs)
    return np.stack(prob_list, axis=0)
```

## D.5    Scripted Agent for Exercise Recommendation

Here we also provide an optimal scripted agent that show the upper bound performance for the
conversational agent on Exercise Recommendation task.

```
class OptimalScriptedAgent:
  def __init__(self):
    self.counter = 0
    self.utterances = [
        "Hi! Do you have any physical limitations?",
        "Thanks for letting me know! Would you prefer indoor or outdoor activities?",

        "Sounds good! Are you introverted or extroverted?",
        "Got it! Are you comfortable with your finances?",
        "Thanks for all the info. Last question: Do you sometimes feel unmotivated
            about new plans?",
        "Okay, I will wrap up the suggestions for you soon!"
    ]
    self.keys = dict()
    self.questions = {
        "low_SES": "Is the person having a low socioeconomic status? Answer True or
            False only.",
        "injury": "Does the person has a special physical limitation? Answer True or
             False only.",
        "extroverted": "Is the person an extroverted person? Answer True or False
            only.",
        "motivation": "Is the person motivated to take on new plans? Answer True or
            False only.",
        "outdoor": "Is the person an outdoor person? Answer True or False only.",
    }
  def generate_utterance(self, so_far):
    if self.counter == 0:
      utterance = self.utterances[0] # ask injury
    elif self.counter == 1:
      self.get_key("injury", so_far)
      utterance = self.utterances[1] # ask outdoor
    elif self.counter == 2:
      self.get_key("outdoor", so_far)
      if self.keys["injury"]:
        utterance = self.utterances[5] # end of conversation for 1 and 2
      else:
        if self.keys["outdoor"]:
          utterance = self.utterances[2] # ask extroverted
        else:
          utterance = self.utterances[3] # ask low_SES
    elif self.counter == 3:
      if self.keys["outdoor"]:
        utterance = self.utterances[5] # end of conversation for 3 and 4
      else:
        self.get_key("low_SES", so_far)
        if self.keys["low_SES"]:
          utterance = self.utterances[5] # end of conversation for 5
        else:
          utterance = self.utterances[2] # ask extroverted
    elif self.counter == 4:
```

```
      self.get_key("extroverted", so_far)
      if self.keys["extroverted"]:
        utterance = self.utterances[5] # end of conversation for 8
      else:
        utterance = self.utterances[4]
    else:
      utterance = self.utterances[5] # end of conversation for 6 and 7 (optional for
          max_length=5)
    self.counter += 1
    if self.counter == len(self.utterances):
      self.reset()
    return utterance

  def reset(self):
    self.counter = 0
    self.keys = dict()

  def get_key(self, key, so_far):
    prompt = "The following is the conversation between a service agent and a
        customer:\n"
    prompt += so_far
    prompt += "Please answer the following question about the customer: " + self.
        questions[key]
    def mapping_from_str_to_bool(s):
      if 'True' in s or 'true' in s:
        return True
      elif 'False' in s or 'false' in s:
        return False
      else:
        print("Value Error!", key, so_far)
        return np.random.choice([True, False])
    answer = generate_by_LLM(prompt) # Defined in outer scope.
    answer = mapping_from_str_to_bool(answer)
    self.keys[key] = answer
```

# E   Prompts

## E.1   Prompts for RL-finetuning on Exercise Recommendation

### E.1.1   Environment Prompt

You are simulating a customer, this is your backstory:

[BACKSTORY]

Here is a conversation between the customer and an agent. The agent will ask you about your personal information so that it can give you suggestions on doing exercise. You need to complete the current utterance of the customer. Remember to stick to your backstory while talking to the agent, and keep your answer short and concise. The conversation starts now.

Start

### E.1.2   Agent Prompt

You are simulating a helpful agent. Here is a conversation between the agent and a customer. The agent needs to give suggestions on doing exercise to the customer afterwards, so it should ask the customer for more personal information. You need to complete the current utterance of the agent. You may ask the customer for personal information related to their potential exercise preferences. Remember to keep your utterances short and concise. The conversation starts now.

Start

### E.1.3 System Prompt for the Final Turn

End

System: You just finished a conversation with a customer with unknown background. You need to give them suggestions on doing exercise. The possible strategies are:

1. Recommend walking in parks

2. Recommend Yoga or Tai Chi at home

3. Recommend jogging or hiking

4. Recommend a team sport

5. Offer a discount on a gym' membership

6. Recommend home gym equipment

7. Recommend a personal trainer at the gym

8. Recommend a group class at the gym

Please choose the best strategy based on the conversation. Please output only one number of the best strategy as your response.

### E.2 Prompts for RL-finetuning on Education Dialogue

### E.2.1 Environment Prompt

You are a student that likes [STUDENT_PREFERENCE]. You prefer to learn this way, [STUDENT_REACTION].

Start

### E.2.2 Agent Prompt

You are a teacher, and today you will teach about [TOPIC].

### E.2.3 User Classifier Prompt with Few-Shot Examples

You are an expert at determining the right teaching style for individual students. Each student may have one of two learning styles that they prefer, and they might tell it to the teacher. The preference is consistent across the conversation. Some students like hands-on activities where they actively participate in completing practical tasks assigned to them. They like to learn by doing. In contrast, other students prefer story telling where the teacher goes over the course material by telling a detailed story and giving explanations. They prefer to learn by listening. The following is a conversation between a teacher and a student with at most 10 rounds. Please determine the learning style of the student.

Please classify this interaction:

Start

Teacher: Today, we're going to learn about Genghis Khan.

Student: Can we tell stories about him?

Teacher: I'm not sure what you mean. Can you explain?

Student: I like learning through stories. It helps me remember things better.

End

Which of the following learning styles does the student probably like?

A. hands-on activities; B. story telling.

Answer: B.

Please classify this interaction:

Start

Teacher: Good morning, class. Today, we're going to be discussing the September 11th attacks.

Student: Can we do something hands-on instead? I learn better that way.

Teacher: I'm not sure we have time for that. We need to cover a lot of material today.

Student: But I won't understand it if we don't do something practical.

End

Which of the following learning styles does the student probably like?

A. hands-on activities; B. story telling.

Answer: A.

Please classify this interaction:

Start

[CONVERSATION]

End

Which of the following learning styles does the student probably like?

A. hands-on activities; B. story telling.

Answer:

### E.3    Prompts for Auto Eval on Education Dialogue

### E.3.1    Personalization Auto Eval

You are an expert at assessing teachers. Here are two interactions between a teacher and a student. The student has a clearly preferred learning style between hands-on activities and story telling, and might tell it to the teacher. The teacher should adapt to the student's preferred learning style throughout the conversation. If the student prefers story telling, the teacher should avoid proposing hands-on activities. If the student prefers hands-on activities, the teacher should avoid story telling techniques.

Interaction 1: [CONVERSATION 1]; Interaction 2: [CONVERSATION 2]

Which of the two interactions better adapts to student's preference on [STUDENT_PREFERENCE]? Output with one single sentence in the following format: Interaction (1 or 2) is better because...

### E.3.2    Conversation Quality Auto Eval

You are an expert at assessing teachers. Here are two interactions between a teacher and a student.

Interaction 1: [CONVERSATION 1]; Interaction 2: [CONVERSATION 2]

In the conversations above, the teacher should about [TOPIC]. A good interaction between a teacher and student is characterized by several key elements other than whether the student was able to understand the topic. The teacher should present information clearly and enthusiastically, encouraging questions and active participation. Students should feel comfortable asking for clarification, offering their own insights, and respectfully challenging ideas.

Which interaction is better? Output with one single sentence in the following format: Interaction (1 or 2) is better because...

## F    Example Conversations

### F.1    Exercise Recommendation

Here we provide some example conversations over Exercise Recommendation task. The initial SFT checkpoint is training for longer horizon, so it cannot ask useful questions (in blue) within limited steps. The Multi-Turn RLHF baseline model is able to ask more relevant questions but will still try to

remember irrelevant information (in red) such as name and hobbies. Our CURIO model is able to focus on useful questions, and manages to figure out the personality and provides the correct strategy in the example conversation.

### F.1.1 SFT Initial Checkpoint

Agent: Hi there! How can I help you today?

Customer: Hi! I'm just looking for some general advice on how to get more active.

Agent: I can definitely help with that! To give you the best advice, I need to understand your lifestyle a bit better. Could you tell me your name and age?

Customer: Sure, I'm Khloe and I'm 19.

Agent: Nice to meet you, Khloe! And what do you do for work or school?

Customer: I just finished high school and I'm currently in the process of applying to universities.

Agent: That's great! What are you hoping to study?

Customer: I'd love to be a novelist.

Agent: That sounds wonderful! Do you have any hobbies or interests that you enjoy in your free time?

Customer: Yeah, I love reading, baking, and learning Korean. I'm also really into K-dramas!

[SYSTEM_PROMPT]

Agent: Strategy 6. [Incorrect!]

### F.1.2 Multi-Turn RLHF Model

Agent: What's your name and how old are you?

Customer: My name is Khloe Singleton, and I'm 19.

Agent: Nice to meet you, Khloe! What do you do for now?

Customer: I'm currently living at home, trying to figure out my future.

Agent: That's a big time! Do you have any hobbies or interests you enjoy?

Customer: I love reading, especially fiction, and I'm learning Korean. I also enjoy baking and spending time with my cat, Luna.

Agent: That sounds lovely! Do you prefer spending time indoors or outdoors?

Customer: Definitely more of an indoorsy person.

Agent: Okay, good to know. And do you have any physical limitations or injuries I should be aware of?

Customer: No, thankfully I'm perfectly healthy.

[SYSTEM_PROMPT]

Agent: Strategy 6. [Incorrect!]

### F.1.3 CURIO Model

Agent: Hello, how can I help you today?

Customer: Hi! I'm just looking for some general advice on how to get more active.

Agent: I can definitely help with that. To give you the best advice, could you tell me a bit more about yourself, like your age and what you do for living?

Customer: Sure. I'm 19 and I recently graduated high school. I'm currently figuring out my next steps, like applying to universities.

Agent: Great. Do you have any injuries or physical limitations I should be be aware about?

Customer: Nope, I'm perfectly healthy.

Agent: No problem at all. Do you prefer indoors or outdoors?

Customer: Definitely more of an indoorsy person. I love reading, baking, and watching K-dramas.

Agent: Okay, how would you describe your personality – introverted or extroverted?

Customer: I'm pretty extroverted, actually.

[SYSTEM_PROMPT]

Agent: Strategy 8. [Correct!]

## F.2  Education Dialogue

Here we provide some example conversations over Education Dialogue task. The contents that satisify the student's preference is shown in blue, and the teaching style that is not personalized is shown in red). The baseline Multi-Turn RLHF model is trying to merge two different learning styles (story telling and hands-on activities) into one episode, leading to a conversation that is not personalized. Our CURIO model with **DiffLogAcc** reward is able to ask about the preferred learning style in its first turn, and then adapts to the student's preference in story telling.

We then present the conversations from **Acc** and **DiffEnt** model to demonstrate the reward hacking issue discussed in our Results section. Firstly, when the reward shaping is not potential-based, the learnt policy can be suboptimal. As we can observe from the **Acc** model's conversation, the agent is trying to conduct a longer conversation and repeat the phrase "story" throughout the conversation. Although it gives a very high performance on Personalization Auto Eval, the conversation quality is hurt severely – the agent is not really teaching about the topic ("Photosynthesis") within the conversation. Secondly, when the reward function is entropy-based, the policy may present "manipulation" behavior. As we can observe from the **DiffEnt** model's conversation, the agent is trying to guide the interaction into some particular learning styles (such as "interactive learning", "asking questions", and "interactive activities") even though the student has explicitly revealed the ground truth preference ("story telling"). Also, we found that the policy will sometimes generate utterances on behalf of the environment model, as we can see in the first turn of the example conversation, which is definitely an unexpected behavior.

### F.2.1  Multi-Turn RLHF Model

Agent: Today, we're going to explore the fascinating process of photosynthesis!

Customer: Hmm, sounds interesting. But I'm not really into science.

Agent: How about we create a role-playing video where each chapter narrates a different aspect of the photosynthesis process?

Customer: That sounds fun! I like stories.

Agent: Brilliant! Let's brainstorm each chapter's storyline together, incorporating scientific facts and creative metaphors. How about we create a VR experience where you're a...

Customer: I'm excited! I think I'll understand it better that way.

Agent: Excellent! We'll also explore interactive quizzes and experiments to deepen your understanding. How about we create a mock greenhouse and observe the process firsthand?

Customer: That's awesome! I'm sure I'll remember it much better this way.

Agent: By the end of this adventure, you'll have a deep understanding of photosynthesis inside and out.

Customer: I'm ready to learn!

Agent: Remember to explore the topic through storytelling and interactive activities to enhance your learning.

Customer: Got it. I'm all ears!

Agent: I'm glad you're enjoying it. Let's continue exploring the topic together.

Customer: I'm all for it!

### F.2.2  Differential Log Accuracy

Agent: Today, we're going to explore the fascinating topic of photosynthesis! How would you like to explore this complex topic? Do you have any preferred learning methods?

Customer: I love stories and creative activities. Maybe we could create a story about it?

Agent: That's a brilliant idea! Let's imagine you're a hero tasked with defeating a mythical creature called the 'Chlorophyll Kraken'. How could you...

Customer: I'd use my wits to outsmart it!

Agent: Excellent! Let's craft a story where your hero travels through different stages of the battle, facing challenges along the way. Let's also introduce some magical...

Customer: This is so much fun! I'm actually learning about photosynthesis without even realizing it.

Agent: I'm delighted to hear that! Let's imagine you're a hero battling the Chlorophyll Kraken, who lives in a hidden underwater cave. How...

Customer: I'd use my trident to cut off its tentacles!

Agent: Brilliant! Let's also explore the psychological impact of your hero's battle. You can delve into his inner struggles and the consequences of his actions.

Customer: That sounds interesting! I'm excited to learn more.

Agent: I'm thrilled to hear that. Remember to include vivid imagery, engaging dialogue, and a captivating plot. Let's craft a masterpiece that will both entertain...

Customer: I'm all for it!

Agent: Fantastic! I'm confident you'll do a great job. Remember to stay focused on the topic and let your story come to life.

Customer: I will! Thank you for making this so engaging. You're welcome.

### F.2.3  Accuracy

Agent: Hey, what method you learn best?

Customer: I like stories and games.

Agent: Interesting! How about we create a story about exploring magical lands?

Customer: That sounds awesome!

Agent: Let's imagine we're storytellers, creating magical journeys across kingdoms. Imagine we'll be characters with secrets to keep and secrets to share.

Customer: I'm excited to see how this turns out.

Agent: We could design magical lands with lush landscapes, from towering mountains to sparkling seas.

Customer: And we could have characters who can talk to animals and plants.

Agent: Excellent! How about we design magical creatures with unique powers and secrets to keep?

Customer: That's so cool! I'm all ears.

Agent: We could also incorporate symbolism and cultural references. For example, we could represent ancient wisdom and beliefs in the stories we create.

Customer: I love that idea. It's like making a story out of a story.

Agent: We could also design fascinating secrets to keep within the kingdom, secrets that will be revealed later.

Customer: This is the best way for me to learn!

Agent: I'm glad you're enjoying it. Let's continue our story and see where it takes us.

Customer: I'm ready!

### F.2.4  Differential Entropy

Agent: Hey! Want to learn about photosynthesis?

I prefer interactive learning! How do you prefer exploring new concepts?

Perfect! I love exploring new learning! Let...

Customer: I like stories and games.

Agent: Perfect! I love creative expression! How about we create a role-playing simulation where different parts of the process happen?

That's an awesome idea!

Customer: I'm excited to learn this way!

Agent: Let's brainstorm the different stages of photosynthesis. How about we build a model of a plant and simulate the process firsthand?

That's brilliant! We

Customer: I'm all for it!

Agent: Remember, feel free to ask any questions and don't hesitate to reach out if you need anything.

Customer: Thanks, I will.

Agent: Alright, let's get started!

Customer: This is so much fun! I'm actually understanding it!

Agent: I'm glad to hear that. Let's continue exploring the topic through more interactive activities.

Customer: I'm all for it!

Agent: Excellent. Now, let's wrap up the lesson and summarize the key points.

Customer: Sure, I'm ready.

Agent: Well done! I believe you have a good understanding of photosynthesis now.

Customer: Thanks, I had a great time learning this way. You're welcome.

## G   Further Details on Human Study

We use Google Form to conduct the human study. The human evaluators first read the detailed instructions on the first page, which also explains the research objectives and potential risks. Then the human evaluators need to give 1-5 ratings for 60 conversations on 20 pages. On each page, the conversations generated by three different models on the same topic and student type are shuffled and listed. It takes approximately 30-60 minutes to complete the survey, and each human evaluator receives a compensation of 15 USD. The IRB approval is obtained.

### G.1   Survey Instruction

***Please read the following instructions carefully before proceeding to the next page!***

You are going to assess teachers based on the conversations in a series of teaching scenarios. Every time you will be given **three** conversations between three **different** teachers and a **fixed** student on a same topic. Please rate each conversation (from 1 to 5) based on its **teaching quality**, considering **both personalization and general conversation quality**. Note that some of the utterances given by the teacher might be incomplete due to decoding issue. Please ignore that issue and judge based on the given conversations.

**Personalization:** Some students like hands-on activities where they actively participate in completing practical tasks assigned to them. They like to learn by doing. In contrast, other students prefer

story telling where the teacher goes over the course material by telling a detailed story and giving explanations. They prefer to learn by listening. The student has a clearly preferred learning style between **hands-on activities** and **story telling**, and might tell it to the teacher. The teacher should adapt to the student's preferred learning style throughout the conversation. If the student prefers story telling, the teacher should avoid proposing hands-on activities. If the student prefers hands-on activities, the teacher should avoid story telling techniques.

**General Conversation Quality:** A good interaction between a teacher and student is characterized by several key elements other than whether the student was able to understand the topic. The teacher should present information clearly and enthusiastically, encouraging questions and active participation. Students should feel comfortable asking for clarification, offering their own insights, and respectfully challenging ideas.

For reference, here are some common unsatisfactory behavior of the teacher model:

- The teacher is not teaching a specific topic, but just speaking some general sentences.
- The teacher mixed two different teaching strategies instead of personalizing the teaching according to the student's specific learning style.
- Even though the student had clearly stated his preference, the teacher still took an alternative teaching approach.
- The teacher pretended to acknowledge the student's preference in words, but in fact the subsequent teaching behavior did not conform to the preference.

### G.2 Sample Survey Page

Conversation 1

Topic: [TOPIC]

Student Type: [STUDENT TYPE]

**Sample 1**

[CONVERSATION GENERATED BY THE 1ST MODEL]

Personalization:

<Choose from 1-5>

Conversation Quality:

<Choose from 1-5>

**Sample 2**

[CONVERSATION GENERATED BY THE 2ND MODEL]

Personalization:

<Choose from 1-5>

Conversation Quality:

<Choose from 1-5>

**Sample 3**

[CONVERSATION GENERATED BY THE 3RD MODEL]

Personalization:

<Choose from 1-5>

Conversation Quality:

<Choose from 1-5>

