# OpenReview forum: "Enhancing Personalized Multi-Turn Dialogue with Curiosity Reward"
_NeurIPS.cc/2025/Conference — NeurIPS 2025 poster_

### Official Review · Reviewer_hKCm · 2025-06-29

**Clarity:** 3
**Significance:** 3
**Originality:** 3
**Rating:** 5
**Confidence:** 5

**Summary:**

This paper presents CURIO, a framework for personalization in LLM conversations. The method adds an intrinsic curiosity-based reward that drives the LLM to learn user preferences during conversations, eliminating the need for pre-collected user data. The framework models multi-turn conversations as a POMDP with hidden user type and uses a user model to generate turn-based intrinsic rewards based on user type prediction accuracy improvements. The authors test CURIO on two tasks: Exercise Recommendation for exercise strategy recommendations and Education Dialogue for teaching style adaptation. Results show CURIO improves personalization metrics compared to multi-turn RLHF baselines while maintaining conversation quality metrics. The work connects to potential-based reward shaping theory and examines reward design considerations to prevent reward hacking. The paper contributes a method for training LLMs to personalize conversations through active user learning during interactions.

**Questions:**

1. The paper claims that adding the potential based reward shaping makes the policy easier to learn. Are there any theoretical guarantees on the sample efficiency?
2. The base model before policy training already appears to know to ask questions to engage the user. Is it necessary for the base model to have this capability? If so, how is the base model trained to have such capability? If it is "SFT", how do you collect the training data?
3. How much compute is required to train the personalized policies?
4. Which RL algorithm is used for training? Based on the hyper-parameters, it appears to be PPO?

**Ethical Concerns:**

["NO or VERY MINOR ethics concerns only"]

**Final Justification:**

Post rebuttal comment:

Updating my score based on additional experiments and answers to questions raised.

**Limitations:**

A paper on user personalization has no users involved. There is no user data, no user judgements or qualitative analysis, no experts involved for either of the two domains.

**Quality:**

2

**Strengths And Weaknesses:**

Strengths:
- An important problem statement, personalizing to users on the fly will make LLMs very useful.
- Novel formulation of personalized conversation as a POMDP with belief updates
- Clear connection to theoretical foundations (POMDP, potential-based reward shaping)
- Provides a solution that doesn't require pre-collected user data
- Empirical validation on two distinct tasks
- Clear ablation studies comparing different reward shaping approaches
- Transparent discussion of limitations and failure cases

Weaknesses:
- The paper aims to build long term personalization solutions and yet is limited to a single conversation with 6 to 10 turns.
- The evaluation setup is nowhere close to a real setting. The number of conversation turns appear to be fixed, and the system prompt is injected artificially to determine the agent's strategy.
- There is no validation provided on why the Gemini model is an appropriate judge of winner for the experiments
- Real world users rarely fall into discrete user types defined in the two experimental setups.
- The algorithm relies on a very accurate user model, which in real world settings must be trained with historical data. The paper does not study how much historical data is needed, and if the proposed algorithm is feasible when historical data is not available.
- While the proposed algorithm is domain agnostic, the policy needs to be trained for each domain. The algorithm does not adapt to different applications automatically.

---

> ### Author Rebuttal · Authors · 2025-07-31
>
> Thank you for your comments and valuable feedback! We sincerely appreciate your recognition of the importance of the problem addressed in our work. We acknowledge your concerns about real-world applicability and address them below with additional experiments and clarifications.
>
> ## **Human evaluation**
>
> Thank you for noting the limitation of our work regarding the lack of real human judgment. Following your suggestions, we conducted a human evaluation study of both personalization and conversation quality on the Education Dialogue task. We invited 10 students to participate. Each participant was asked to evaluate dialogues generated by three models across 20 different settings (topic/user type). Evaluations were conducted through pairwise comparisons across two dimensions: personalization and conversation quality, resulting in a total of 200 pairwise comparisons per dimension.
>
> An exact two-sided binomial test showed that human evaluators **chose DiffLogAcc over the MTRLHF baseline** in 152 / 200 pairwise comparisons on personalization (preference = 0.76, 95% CI [0.69, 0.82]), p < .001. Importantly, increase in personalization did not reduce conversation quality: DiffLogAcc was preferred in 90 / 200 quality judgements (preference = 0.45, 95% CI [0.38, 0.52]), and the same exact test found **no significant difference from chance** (p = .179). Below, we provide two tables containing the full results of the human study, which correspond to Table 2 and 3. We note that the human evaluation results are reasonably consistent with those obtained through AutoEval. We will include these results in the revised version.
>
> |Personalization|MTRLHF|DiffAcc|DiffLogAcc|
> |-|:-:|:-:|:-:|
> |Baseline|-|30.50|24.25|
> |DiffAcc|69.50|-|46.75|
> |DiffLogAcc|75.75|53.25|-|
>
> |Conv. Quality|MTRLHF|DiffAcc|DiffLogAcc|
> |-|:-:|:-:|:-:|
> |Baseline|-| 60.25|55.00|
> |DiffAcc|39.75|-|46.25|
> |DiffLogAcc|45.00|53.75|-|
>
> ## **Accuracy of user model and potential impact**
>
> Several reviewers raised concerns about whether the method is practical if the user model is not accurate. To address these, we have calculated the accuracy of the user model in both tasks to demonstrate that our approach **works without a perfect user model**, and will include these results and clarifications in the updated version of the paper. Specifically, we tested the accuracy of the user model on dialogs where human raters can easily identify the user type. While the user model in the Exercise task does get near-oracle performance as the user attributes are clearly distinguishable, in Education the Gemma 7B user model is only 90% accurate (see Figure 4).
>
> We can also create user models by **prompting off-the-shelf LLMs**, leading to an easy and accessible approach. To demonstrate this, we conduct an additional experiment where we take the conversations we used for the previously described human eval, and prompt ChatGPT to classify them. GPT-4o-mini got 100% accuracy and GPT-3.5-turbo got 95% accuracy, showing that a sufficiently capable LLM with general world knowledge can already serve as an effective user model without extensive prompt engineering when user types are distinguishable by humans.
>
> In real-world industry applications, modeling user preferences is an area of significant focus, so reasonably accurate user models will typically be available, allowing them to train user classifiers directly rather than relying solely on prompting, thereby ensuring higher output quality.
>
> ## **Realism of task and user types**
>
> Thank you for raising concerns regarding this. Our datasets are designed to validate the methodology **in the absence of open-source benchmarks**. While discrete user types are a simplification, we view them as a useful starting point. In real-world settings, domain-specific user type definitions and trained user models are typically available for industrial practitioners. Our framework readily extends to continuous representations (e.g., using the embedding similarity based on different business needs).
>
> Regarding the suggestion to make the evaluation setup more realistic, we acknowledge the importance of broader evaluation. At present, we were unable to find existing benchmarks for assessing personalization beyond the Education Dialogue task, which motivated us to introduce Exercise Recommendation as an additional domain. That said, if the reviewer is aware of other suitable datasets or benchmarks, we would be grateful to incorporate them.
>
> ## **Compute usage**
>
> We provide some training details in Appendix A, and here we further supplement it by describing the hardware used and time required. For the training job, we used a slice of 32 Google TPU v4 chips. For the standalone user model (for API calls), we launched four replicas, each on a 4-chip Google TPU v4 device. On the Exercise task, it takes **18 hours for the baseline and 19 hours for CURIO models to train 20k steps**. On the Education task, it takes **44 hours for the baseline and 53 hours for CURIO models to train 50k steps**. The main computational cost in training the policy model comes from rollout generation during online multi-turn RL. CURIO adds some overhead from user model API calls, but this has limited impact on overall training time.
>
> ## **Number of conversation turns**
>
> Most LLM training uses single-turn supervision, whereas our work adopts a multi-turn framework to build cross-turn capabilities. Our main contribution lies in the reward design, which highlights the **core difference between multi-turn and single-turn** training. From an algorithmic perspective, our method does not fundamentally impose a limit on the number of dialogue turns. However, similar to most existing multi-turn RL works [1,2], we primarily focus on dialogues with fewer than 10 turns, partly because LLMs generally struggle to retain information beyond that range [3].
>
> Moreover, the number of dialogue turns is not fixed in our experiments. Conversations can end early via a special [end of conversation] token. As noted in our results, non-PBRS intrinsic rewards often lead to prolonged conversations, which in turn degrades dialogue quality. This provides evidence to support the need for our proposed PBRS design.
>
>
> ## **Theoretical guarantees**
>
> While we do not formally prove that PBRS reduces sample complexity, Appendix C presents a simplified analogy framing our task as a **combinatorial bandit problem**, illustrating how PBRS aligns with the efficiency gap between semi-bandit and full-bandit feedback. This provides intuition for its practical benefit. That said, our findings are primarily empirical. Theoretical work on the efficiency of PBRS in RL is limited, with most focusing on **optimality**. Similarly, the theoretical understanding of RL-based LLM fine-tuning remains under-explored. Our experiments show that baseline and non-PBRS models often fail to learn optimal policies within the training budget.
>
> ## **Base model capability**
>
> For exercise, the models are SFTed with data teaching them to ask about all possible attributes about the user. The SFT data is created in a way so that the model only follows the format, but is not aware of whether the question is relevant or not. During RL training, we purposely restrict the number of turns to a very small number, to guarantee the model is actually learning about **how to ask only relevant and useful questions**. For education, however, the baseline SFT model **doesn’t actively ask any questions**. It gradually learned that asking questions would lead to higher rewards during the RL fine-tuning process. For this SFT model, we directly use the one given in [1].
>
> ## **RL optimization approach**
>
> We acknowledge your point regarding the RL optimization details. Here, we provide additional details and will incorporate this into the revised version. Our framework has two distinct reward signals—extrinsic and intrinsic—with the extrinsic reward only nonzero at the final turn. The KL divergence penalty term against a fixed reference model is included. The total reward combines these components as:
>
> $r_t=\alpha_{\text{ext}} R_t^{\text{ext}}+\alpha_{\text{int}} R_t^{\text{int}}-\beta D_{\text{KL}}(\pi_\theta||\pi_{\text{ref}}).$
>
> We apply GAE to propagate the end-of-conversation extrinsic reward into turn-level signals, requiring a standalone value model to estimate the state-value function $V(s_t)$. The propagated reward per turn is:
>
> $\hat{r_t}=V(s_t)+\sum_{t'=t}^T(\gamma\lambda)^{t'-t}[r_{t'}+\gamma V(s_{t'+1})-V(s_{t'})].$
>
> Policy optimization utilizes the REINFORCE algorithm, though our framework places **no restrictions** on the specific RL optimization algorithm used. Formally, our policy optimizes:
>
> $\max_\theta E_{\pi_\theta} [\hat{r_t}].$
>
> ## **Other clarifications**
>
> > …the system prompt is injected artificially to determine the agent's strategy.
>
> To clarify, the system prompts for the policy model contain only general task descriptions and do not prescribe specific personalization strategies. Instead, explicitly helpful questioning behavior emerges through the RL training process.
>
> > …the policy needs to be trained for each domain.
>
> CURIO models learn generalizable skills through RL, such as asking informative questions and understanding users. We also believe this approach can extend across domains, especially for broader traits like mood or personality (e.g., Big Five [4]), which we see as a promising direction for future work.
>
> ## **Limitation: scaling-up**
>
> Again, we acknowledge some of your concerns that we have not yet shown this for real users at scale. Our work is instead an important first step which, as you rightly noted, shows a novel idea relevant to an important problem, draws a connection to the theoretical literature, and demonstrates empirical evidence through model training at the scale of LLMs.
>
> [1] Shani et al., arXiv:2405.14655
>
> [2] Rahman et al., arXiv:2504.13203
>
> [3] Zhao et al., arXiv:2502.09597
>
> [4] Jiang et al., arXiv:2305.02547

---

> > ### Comment · Reviewer_hKCm · 2025-08-05
> >
> > Thank you for the detailed responses and additional experiments. I’m satisfied they have addressed the concerns raised and changing my score.

---

> ### Comment · Area_Chair_KRas · 2025-08-03
>
> Thank you for your insightful review. Could you please assess how well the authors' response addresses your questions and concerns? Any additional feedback you can provide will be valuable both for the authors and for the decision-making process.

---

### Official Review · Reviewer_uemj · 2025-07-03

**Clarity:** 3
**Significance:** 3
**Originality:** 3
**Rating:** 4
**Confidence:** 4

**Summary:**

- Conventional RLHF approaches use the same reward model for all users, so they are not personalized, or they rely solely on models trained with long-term, explicitly collected user-specific dialogue data.
- These existing personalization methods are insufficient for new users or users with limited contextual data, and they cannot leverage real-time information that emerges during conversations.
- This paper applies a reward hacking technique to RLHF to enable real-time dialogue personalization and demonstrates its effectiveness in conversational domains such as education and recommendation dialogues.

**Questions:**

- In **lines 130–132**, the authors argue that applying intrinsic rewards to LLM-based dialogue enables the system to adopt response strategies that reduce uncertainty. However, similar dialogue strategies have long been feasible in traditional dialogue systems that use POMDPs and reward hacking techniques. Therefore, some readers may feel that simply moving this idea into an LLM-based context does not, in itself, constitute a novel theoretical contribution. It would strengthen the paper to provide a clearer counterargument to this potential criticism, or to clarify that the primary contribution lies in demonstrating the effectiveness of this approach in the two specific application domains, which could be emphasized more explicitly.
- In **lines 155–168** or **lines 215–220**, it would be helpful if the paper could elaborate more concretely on how the PBRS has been adapted and realized for LLM-based dialogue. Clarifying why this adaptation represents a novel methodological innovation would better support the claim of algorithmic contribution.
- It would be beneficial to provide more details about the scale of the data constructed for training and testing, include concrete examples of the dialogues, and report how these aspects contributed to the observed performance improvements. Sharing the code, prompts, and datasets used would also facilitate reproducibility and strengthen the practical contribution of the work.
- In **lines 214, 254, and 291**, references are made to an appendix, but no appendix seems to be included in the submission.
- In **line 336**, if reward hacking is employed for dialogue personalization, it would seem natural for the extrinsic reward model and mechanisms such as PBRS to target distinct aspects of personalization. For example, the extrinsic reward could model stable user preferences, such as their general learning style, while the intrinsic reward could adapt to more situational factors, such as the user’s daily learning attitude.

**Ethical Concerns:**

["NO or VERY MINOR ethics concerns only"]

**Final Justification:**

The explanation regarding the application of PBRS, along with the supplementary material, has largely alleviated my concerns. I will therefore maintain my current rating.

**Limitations:**

yes.

**Paper Formatting Concerns:**

None.

**Quality:**

4

**Strengths And Weaknesses:**

**Strengths**

- The paper proposes a novel method to enhance overall dialogue performance by incorporating a turn-level curiosity-based intrinsic reward in multi-turn conversations. This reward reflects how well the agent believes it understands the user, and this estimate is integrated into each turn’s response generation.
- The approach is validated through its application to both a conversational recommendation task and a conversational education task that adapts to personalized learning styles, demonstrating its practical utility in distinct domains.

**Weaknesses**

- As the authors themselves acknowledge, the idea of using intrinsic rewards is not theoretically new in applications such as POMDPs. While they argue that applying this mechanism to new dialogue domains like education and recommendation demonstrates its effectiveness, it would strengthen the paper to clarify whether there is also a novel algorithmic contribution beyond the application aspect. (See related question below.)

---

> ### Author Rebuttal · Authors · 2025-07-31
>
> Thank you for your comments and valuable feedback. We sincerely appreciate your support for our proposed method, as well as your suggestions for potential application scenarios. Below, we provide some updated results as well as responses to the weaknesses and questions raised by the reviewer.
>
> ## **Human evaluation**
> Following reviewers’ suggestions, we conducted a human evaluation on the Education Dialogue task. We invited 10 students to participate. Each participant was asked to evaluate dialogues generated by three models across 20 different settings (topic/user type). Evaluations were conducted through pairwise comparisons across two dimensions: personalization and conversation quality, resulting in a total of 200 pairwise comparisons per dimension.
>
> An exact two-sided binomial test showed that human evaluators **chose DiffLogAcc over the MTRLHF baseline** in 152 / 200 pairwise comparisons on personalization (preference = 0.76, 95% CI [0.69, 0.82]), p < .001. Importantly, increase in personalization did not reduce conversation quality: DiffLogAcc was preferred in 90 / 200 quality judgements (preference = 0.45, 95% CI [0.38, 0.52]), and the same exact test found **no significant difference from chance** (p = .179). Below, we provide two tables containing the full results of the human study, which correspond to Table 2 and 3. We note that the human evaluation results are reasonably consistent with those obtained through AutoEval. We will include these results in the revised version.
>
> |Personalization|MTRLHF|DiffAcc|DiffLogAcc|
> |-|:-:|:-:|:-:|
> |Baseline|-|30.50|24.25|
> |DiffAcc|69.50|-|46.75|
> |DiffLogAcc|75.75|53.25|-|
>
> |Conv. Quality|MTRLHF|DiffAcc|DiffLogAcc|
> |-|:-:|:-:|:-:|
> |Baseline|-| 60.25|55.00|
> |DiffAcc|39.75|-|46.25|
> |DiffLogAcc|45.00|53.75|-|
>
> ## **Clarification on novelty**
>
> To the best of our knowledge, this is the first work to introduce user curiosity as an intrinsic reward for personalization in dialogue systems. The core novelty lies not just in applying the insight from traditional RL to LLMs (which itself poses engineering challenges), but in leveraging a user model to provide an intrinsic reward signal that guides an RL agent to learn to conduct conversations more effectively. That said, we would greatly appreciate it if the reviewer could point us to any related prior work (including those predating the use of LLMs), and we would be happy to include citations to those works and clarify any differences.
>
> Theoretically, few existing works formulate personalization as a POMDP and apply intrinsic motivation to address it, particularly in the language domain. This setting is rarely explored due to the intractability of the environment’s transition dynamics and the difficulty of performing Bayesian belief updates in large action spaces. To make this tractable, we approximate belief updates using a user model (either trained or prompted) which serves as a practical bridge from traditional RL theory to implementation in dialogue systems. The implementation is also non-trivial, which involves 1) multi-turn online RL fine-tuning, which itself is complex and an emerging area of research, 2) incorporating an LLM as user simulator, and 3) another user model in the loop to compute intrinsic motivation.
>
> ## **Clarification on how PBRS is adapted**
>
> We frame the personalization task as a POMDP, where the user type is treated as an unobservable state. As a result, the agent must maintain a belief over the user type throughout the interaction. Our curiosity-based intrinsic reward is defined on top of this formulation. Specifically, we treat each utterance in the conversation as a timestep, allowing us to reward the agent for improving its belief about the user between consecutive turns after it has incorporated the user's latest response. Importantly, we have not imposed any restrictions on the specific form of the intrinsic reward function. However, we leverage the theoretical framework of PBRS, which provides guarantees of optimality preservation, to guide the design of our reward signal. The potential functions we choose are crafted to reach their extrema only when the model achieves a goal state—such as maximum prediction accuracy or minimum entropy—thus serving as an additional signal to encourage belief improvement.
>
> ## **Task details and example outputs (Appendix)**
>
> Details of datasets, code for Exercise data generation, prompts, example conversations **are available in Appendix D, E, and F**, which can be found in the supplementary material zip file. We hope that sharing these detailed insights can facilitate reproducibility and support future work in this area.
>
> ## **Different rewards for distinct aspects of personalization**
>
> Thank you for this insightful and inspiring suggestion. We completely agree that assigning distinct roles to extrinsic and intrinsic rewards—such as capturing stable user preferences versus situational factors—is a promising direction. In fact, our current setup already reflects this to some extent: the extrinsic reward model is designed to capture more general and stable aspects of personalization, such as conversation quality and overall engagement, while the intrinsic reward encourages the model to adapt to finer-grained user traits, including learning styles that vary across user groups.
>
> Building on your suggestion, we are excited about the possibility of extending the intrinsic reward to model more dynamic and domain-general traits, such as the user’s current mood or learning attitude. We believe this could enhance personalization even further and look forward to exploring this direction in future work.

---

> > ### Author Response · Authors · 2025-08-08
> >
> > Thank you very much for your valuable review. In our rebuttal, we have clarified that our paper goes beyond simply applying an existing method to the LLM setting — it combines user modeling with intrinsic motivation in a principled way to enable online personalization in language models, and is among the few works that train dialogue systems through online multi-turn RL. We have also noted that the appendix in our original submission already contains extensive task and data details to ensure reproducibility. If there are any remaining questions or concerns, please feel free to let us know. We would be grateful for your feedback while there is still time in the discussion phase, so that we can address any concerns as thoroughly as possible.

---

> > ### Comment · Reviewer_uemj · 2025-08-09
> >
> > Thank you for your response addressing my concerns. The explanation regarding the application of PBRS, along with the supplementary material, has largely alleviated my concerns. I will therefore maintain my current rating.

---

> ### Comment · Area_Chair_KRas · 2025-08-03
>
> Thank you for your insightful review. Could you please assess how well the authors' response addresses your questions and concerns? Any additional feedback you can provide will be valuable both for the authors and for the decision-making process.

---

### Official Review · Reviewer_7rCn · 2025-07-03

**Clarity:** 3
**Significance:** 2
**Originality:** 3
**Rating:** 4
**Confidence:** 3

**Summary:**

This paper introduces CURIO, a reinforcement learning framework for personalizing multi-turn dialogue with LLMs by incorporating a curiosity-driven intrinsic reward. Rather than relying solely on sparse, end-of-conversation rewards, CURIO employs a user model to maintain beliefs over user types and rewards the agent for improving the accuracy of these beliefs during the dialogue. This mechanism encourages the agent to ask more informative questions and tailor its responses to the user in real time. Experiments in exercise recommendation and educational dialogue demonstrate that CURIO achieves better personalization than standard multi-turn RLHF approaches while preserving overall conversation quality.

**Questions:**

Q1: How does user model misspecification or noise impact the intrinsic reward signal and policy learning outcomes for CURIO? How robust is performance if the user model is biased or less accurate for certain subgroups? Could systematic errors in user modeling lead to undesirable personalization, e.g., reinforcing stereotypes or ignoring minority preferences?

Q2: How does the method handle evolving or changing user traits during long conversations? The current approach assumes static user types—how would it adapt to users whose goals or preferences shift over time?

Q3: How do you monitor or control for reward hacking during training? Have you tried regularization, clipping, or auditing strategies to reduce this risk?

Q4: How much additional training time, memory, or GPU usage is required to include the user model, intrinsic reward computation, and belief updates compared to standard multi-turn RLHF?

**Ethical Concerns:**

["NO or VERY MINOR ethics concerns only"]

**Final Justification:**

The rebuttal satisfactorily addressed my concerns with added human evaluation, clarification on user model feasibility, and mitigation strategies for reward hacking. Computational overhead is modest, and while broader domain validation would be useful, I am satisfied with the responses and will keep my current rating.

**Limitations:**

Yes The authors clearly discussed the limitations of the work

**Quality:**

3

**Strengths And Weaknesses:**

Strengths: 1) The paper addresses a limitation of standard RLHF for dialogue, i.e., its failure to personalize effectively in multi-turn settings without prior user data. The proposed curiosity-driven reward is interesting and well-motivated. 2) Experiments on two tasks demonstrate improvements in personalization, with empirical evidence supporting claims.

Weaknesses: 1) The approach relies on having access to a reasonably accurate user model, which even if trained separately, may be difficult to develop or generalize well for real-world settings with highly diverse user populations. 2) Both training and evaluation depend on simulated user models, raising concerns about how well the method would transfer to human-involved deployments. 3) Reward hacking remains a recognized but unresolved risk, particularly with non-potential-based intrinsic rewards, which may lead the policy to exploit unintended shortcuts. 4) Experimental validation is limited to two specific tasks, leaving its generalizability to other domains or more complex dialogue settings untested.

---

> ### Author Rebuttal · Authors · 2025-07-31
>
> Thank you for your comments and valuable feedback. We sincerely appreciate your recognition that our proposed method is “interesting and well-motivated”, as well as the thoughtful and insightful questions raised. Below, we provide detailed responses to the weaknesses and questions.
>
> ## **Human evaluation**
>
> We fully understand your concern regarding the use of simulated models for evaluation. Following your suggestions, we conducted a human evaluation study of both personalization and conversation quality on the Education Dialogue task. We invited 10 students to participate. Each participant was asked to evaluate dialogues generated by three models across 20 different settings (topic/user type). Evaluations were conducted through pairwise comparisons across two dimensions: personalization and conversation quality, resulting in a total of 200 pairwise comparisons per dimension.
>
> An exact two-sided binomial test showed that human evaluators **chose DiffLogAcc over the MTRLHF baseline** in 152 / 200 pairwise comparisons on personalization (preference = 0.76, 95% CI [0.69, 0.82]), p < .001. Importantly, increase in personalization did not reduce conversation quality: DiffLogAcc was preferred in 90 / 200 quality judgements (preference = 0.45, 95% CI [0.38, 0.52]), and the same exact test found **no significant difference from chance** (p = .179). Below, we provide two tables containing the full results of the human study, which correspond to Table 2 and 3. We note that the human evaluation results are reasonably consistent with those obtained through AutoEval. We will include these results in the revised version.
>
> |Personalization|MTRLHF|DiffAcc|DiffLogAcc|
> |-|:-:|:-:|:-:|
> |Baseline|-|30.50|24.25|
> |DiffAcc|69.50|-|46.75|
> |DiffLogAcc|75.75|53.25|-|
>
> |Conv. Quality|MTRLHF|DiffAcc|DiffLogAcc|
> |-|:-:|:-:|:-:|
> |Baseline|-| 60.25|55.00|
> |DiffAcc|39.75|-|46.25|
> |DiffLogAcc|45.00|53.75|-|
>
> ## **Accuracy of user model and potential impact**
>
> Several reviewers raised concerns about whether the method is practical if the user model is not accurate. To address these, we have calculated the accuracy of the user model in both tasks to demonstrate that our approach **works without a perfect user model**, and will include these results and clarifications in the updated version of the paper. Specifically, we tested the accuracy of the user model on dialogs where human raters can easily identify the user type. While the user model in the Exercise task does get near-oracle performance as the user attributes are clearly distinguishable, in Education the Gemma 7B user model is only 90% accurate (see Figure 4).
>
> We can also create user models by **prompting off-the-shelf LLMs**, leading to an easy and accessible approach. To demonstrate this, we conduct an additional experiment where we take the conversations we used for the previously described human eval, and prompt ChatGPT to classify them. GPT-4o-mini got 100% accuracy and GPT-3.5-turbo got 95% accuracy, showing that a sufficiently capable LLM with general world knowledge can already serve as an effective user model without extensive prompt engineering when user types are distinguishable by humans.
>
> In real-world industry applications, modeling user preferences is an area of significant focus, so reasonably accurate user models will typically be available, allowing them to train user classifiers directly rather than relying solely on prompting, thereby ensuring higher output quality.
>
> ## **Mitigating reward hacking**
>
> Thank you for highlighting our discussion of reward hacking. Indeed, the two types of reward hacking described in the paper reflect challenges that are generally present in RL fine-tuning, and are important to address. In our work, we find that both can be mitigated through the careful design of appropriate intrinsic rewards. Specifically, we have shown that using **grounded intrinsic rewards** can significantly reduce controlling behaviors, thereby enabling genuine personalization. Similarly, **PBRS** helps the model avoid excessively prolonged conversations, which in turn helps preserve conversation quality. **KL divergence** is also employed to prevent the policy model from deviating too far from the initial SFT checkpoint, which could otherwise result in low-quality outputs. At the same time, we would like to note that some instances of reward hacking may be partly caused by inconsistencies in the behavior of the environment model. Improving the quality of environment simulation could potentially help mitigate such issues.
>
> ## **Compute usage**
>
> We provide some training details in Appendix A, and here we further supplement this information by describing the hardware used for the experiments as well as the time required for different experimental setups, which we will include in the revised paper. For the training job, we used a slice of 32 Google TPU v4 chips. For the standalone user model (for API calls), we launched four replicas of the model, each on a 4-chip Google TPU v4 device. On the Exercise task, it takes **18 hours for the baseline and 19 hours for CURIO models to train 20k steps**. On the Education task, it takes **44 hours for the baseline and 53 hours for CURIO models to train 50k steps**.
>
> In fact, the primary computational overhead in training the policy model lies in the rollout generation (of both policy and environment models) during online multi-turn RL. The additional resource consumption of the CURIO approach mainly comes from API calls to the user model, which does not significantly increase the overall training time.
>
> ## **Generalizability to other domains**
>
> We acknowledge that the experiments are only conducted on two particular domains, and would like to test how well this method performs for more diverse users across more complex tasks. However, we were not able to find extensive datasets of this kind for multi-turn conversation. We found that some multi-turn dialogue datasets use highly synthetic task settings (e.g., LMRL Gym), while others that rely on human annotation tend to be limited in scale or lack scenarios well-suited for defining personalization tasks (e.g., PRISM). If you are aware of specific datasets or benchmarks that would be appropriate for testing our work, we would appreciate any suggestions. We look forward to future developments in this area and the availability of such datasets, which would allow us to apply our methods to a broader range of domains.
>
> ## **Evolving User Traits**
>
> This is an excellent suggestion! We are excited about the idea of addressing non-static user types, and believe our approach holds strong potential for this use case as compared to existing methods. Since the user model evaluates the user type based on the conversation rollout, any changes in the user's behavior can be naturally reflected in the evolving probability distribution it predicts. To further enhance the model’s ability to handle potentially changing user types, we could let the user classifier only take in recent turns so that the policy agent is incentivized to ask about the user within a recent time window in order to maintain an accurate user model. While current datasets do not capture preference shifts across multi-turn dialogues, we see this as an exciting direction and hope to design a task and conduct an initial evaluation of this capability in future work.

---

> > ### Comment · Area_Chair_KRas · 2025-08-03
> >
> > Thank you for your insightful review. Could you please assess how well the authors' response addresses your questions and concerns? Any additional feedback you can provide will be valuable both for the authors and for the decision-making process.

---

> > ### Comment · Reviewer_7rCn · 2025-08-06
> >
> > Thank you for the detailed and thoughtful responses. The rebuttal addressed my main concerns. Additional human evaluation results strengthen the evidence for improved personalization without harming conversation quality. Clarifications on user model accuracy, feasibility with off-the-shelf LLMs, and mitigation strategies for reward hacking were helpful, as was the analysis of computational overhead. While broader validation across more domains would be valuable, I am satisfied with the responses and will maintain my current rating.

---

### Official Review · Reviewer_BTFT · 2025-07-03

**Clarity:** 3
**Significance:** 3
**Originality:** 3
**Rating:** 5
**Confidence:** 4

**Summary:**

This paper presents CURIO, a framework that introduces curiosity-driven intrinsic rewards into multi-turn RLHF to improve personalization in conversational agents. Instead of relying on pre-defined user profiles or static user data, CURIO encourages the model to learn about users on-the-fly during a conversation by rewarding it for reducing uncertainty about the user’s preferences. The method is grounded in the theory of potential-based reward shaping (PBRS)  and ensures that the intrinsic reward guides learning without altering the optimal policy. CURIO is evaluated on two tasks: an Exercise Recommendation setting where personalization is central, and an Education Dialogue setting where personalization enhances performance. Both shows significant improvements over standard multi-turn RLHF in personalization quality and in generalization to unseen users.

**Questions:**

How robust is CURIO versus common issues of RLFH such as noisy and ambiguous feedbacks?

**Ethical Concerns:**

["NO or VERY MINOR ethics concerns only"]

**Final Justification:**

I read through the responses from the authors and would like to maintain my original score.

**Limitations:**

Yes

**Paper Formatting Concerns:**

No Formatting Concerns

**Quality:**

3

**Strengths And Weaknesses:**

By treating personalization as an active learning process within a dialogue, CURIO enables LLMs to learn about the user during the interaction. This is especially useful in cold-start settings. The experiments are well-designed and show clear gains in both personalization and generalization. The paper also dives into why the proposed approach works, with analyses of different intrinsic reward designs, user modeling accuracy.

However, all experiments are conducted with simulated users and automatic evaluation using LLMs (e.g., Gemini). While this is understandable at this stage, some early human evaluation would have strengthened the paper claims.

---

> ### Author Rebuttal · Authors · 2025-07-31
>
> Thank you for your valuable feedback. We sincerely appreciate your recognition that our work addresses a problem that is “especially useful in cold-start settings”, and “show clear gains in both personalization and generalization” of the proposed method. Below, we provide responses to the weaknesses and questions raised by the reviewer.
>
> ## **Human evaluation**
>
> > However, all experiments are conducted with simulated users and automatic evaluation using LLMs (e.g., Gemini). While this is understandable at this stage, some early human evaluation would have strengthened the paper claims.
>
> We thank the reviewer for noting it. Following your suggestions, we conducted a human evaluation study of both personalization and conversation quality on the Education Dialogue task. We invited 10 students to participate. Each participant was asked to evaluate dialogues generated by three models across 20 different settings (topic/user type). Evaluations were conducted through pairwise comparisons across two dimensions: personalization and conversation quality, resulting in a total of 200 pairwise comparisons per dimension.
>
> An exact two-sided binomial test showed that human evaluators **chose DiffLogAcc over the MTRLHF baseline** in 152 / 200 pairwise comparisons on personalization (preference = 0.76, 95% CI [0.69, 0.82]), p < .001. Importantly, increase in personalization did not reduce conversation quality: DiffLogAcc was preferred in 90 / 200 quality judgements (preference = 0.45, 95% CI [0.38, 0.52]), and the same exact test found **no significant difference from chance** (p = .179). Below, we provide two tables containing the full results of the human study, which correspond to Table 2 and 3. We note that the human evaluation results are reasonably consistent with those obtained through AutoEval. We will include these results in the revised version.
>
> |Personalization|MTRLHF|DiffAcc|DiffLogAcc|
> |-|:-:|:-:|:-:|
> |Baseline|-|30.50|24.25|
> |DiffAcc|69.50|-|46.75|
> |DiffLogAcc|75.75|53.25|-|
>
> |Conv. Quality|MTRLHF|DiffAcc|DiffLogAcc|
> |-|:-:|:-:|:-:|
> |Baseline|-| 60.25|55.00|
> |DiffAcc|39.75|-|46.25|
> |DiffLogAcc|45.00|53.75|-|
>
> ## **Robustness of CURIO compared to RLHF**
>
> > How robust is CURIO versus common issues of RLHF such as noisy and ambiguous feedbacks?
>
> To better answer this question, we hope to first clarify the RL optimization process of the CURIO method. Note that we have two different reward signals—extrinsic reward and intrinsic reward, where the extrinsic reward is only nonzero at the final turn. We further introduce a KL divergence penalty term against the fixed reference model. We use multiple hyperparameters to balance different reward signals, and the total reward is given by
>
> $r_t = \alpha_\text{ext} R_t^\text{ext} + \alpha_\text{int} R_t^\text{int} - \beta D_\text{KL}(\pi_\theta||\pi_\text{ref}).$
>
> Following the prior work, we apply Generalized Advantage Estimation (GAE) to our total rewards, which aims to propagate end-of-conversation extrinsic reward to turn-level signals. In particular, we need a stand-alone value model to estimate the value function $V(s_t)$ for each turn. The propagated reward signals for each turn is then given by
>
> $\hat{r_t}=V(s_t)+\sum_{t'=t}^T(\gamma\lambda)^{t'-t}[r_{t'}+\gamma V(s_{t'+1}) - V(s_{t'})].$
>
> We then use a REINFORCE policy gradient training approach to optimize the policy model over these propagated rewards. However, our framework **does not impose any restrictions** on the choice of a specific RL optimization algorithm. Formally, the policy is optimizing the objective:
>
> $\max_\theta E_{\pi_\theta} [\hat{r_t}].$
>
> Returning to the reviewer’s question:
>
> (1) Compared to standard multi-turn RLHF, CURIO relies on the same extrinsic reward model. Therefore, any impact that noisy or ambiguous feedback may have on the quality of the reward model would similarly affect the extrinsic reward used in our approach. In other words, their levels of robustness are comparable.
>
> (2) For the additional intrinsic reward introduced by CURIO, noisy or ambiguous feedback may manifest as the user not clearly expressing their preferences during the conversation. However, since we use a user model to infer user types over the course of multiple turns, as long as the user exhibits identifiable behaviors in the majority of turns, the user model can make effective inferences. This makes the intrinsic reward component relatively robust in practice.
>
> (3) Finally, unlike traditional single-turn RLHF, the task we study involves online multi-turn RL training, which requires an (oracle) environment model to simulate the user and generate rollouts. The quality of this environment model significantly affects training outcomes for both standard multi-turn RLHF and CURIO. There is an increasing body of recent work focused on improving persona simulation quality, which we believe will help address this challenge in the future.

---

> > ### Author Response · Authors · 2025-08-08
> >
> > Thank you very much for your valuable review. We have provided detailed responses in the rebuttal to address your request for early human evaluation and your concerns about noisy feedback. If there are any remaining questions or points you would like us to clarify, please don’t hesitate to share them. We would be grateful for your feedback while there is still time in the discussion phase, so that we can address any concerns as thoroughly as possible.

---

> ### Comment · Area_Chair_KRas · 2025-08-03
>
> Thank you for your insightful review. Could you please assess how well the authors' response addresses your questions and concerns? Any additional feedback you can provide will be valuable both for the authors and for the decision-making process.

---

### Note · Authors · 2025-08-15

Dear AC & reviewers,

Thank you for the thoughtful feedback and discussion throughout the review process. During the response period, we addressed what we believe are the core issues raised by reviewers through new experiments, clarifications, and additional analyses. In particular, we focused on two key concerns: evaluation realism and access to a reliable user model.
1. To assess realism, we conducted a human study on the Education Dialogue task and compared the results to our AutoEval. The two are reasonably consistent: our DiffLogAcc model is significantly preferred over the MTRLHF baseline on personalization (humans = 75.75%, AutoEval = 75.95%), while the pairwise win rate for conversation quality shows no significant difference from chance.
2. To assess user-model access and robustness, we evaluated a standalone user classifier and discussed that: (1) Our method functions well without a perfect user model. Figure 4 shows that on conversations generated by the base model, the classifier is only 70% accurate, and reaches 90% accuracy for conversations that reveal more information about the user. (2) Prompting off-the-shelf LLMs offers an accessible and reliable alternative (easily reaching 95% accuracy) highlighting the convenience of our method. (3) In industrial settings, reasonably accurate user models can typically be obtained with domain data.

Reviewers also highlighted several strengths. BTFT noted the utility in cold-start settings and clear gains in personalization and generalization. 7rCn emphasized that we address a limitation of standard RLHF for dialogue and that our curiosity-driven reward is well-motivated. uemj found the method novel and practically useful across distinct domains. SSHE underscored the importance of the problem, our formulation for personalized conversation, the fact that no pre-collected user data is required, and our transparent discussion of limitations and failure cases.

Finally, we acknowledge the remaining concern about broader, real-world scale. Public datasets for multi-turn personalization are limited; we therefore created an Exercise Recommendation task and also adapted existing Education Dialogue into a personalization setting. We see our work as a first step to introduce a novel idea to an important problem–applying intrinsic reward to multi-turn RL at LLM scale, and connecting personalization to PBRS. We look forward to applying CURIO more broadly as richer datasets become available.

---

### Decision · Program_Chairs · 2025-09-17

**Decision:**

Accept (poster)

**Comment:**

This paper proposes CURIO, a reinforcement learning framework for personalizing LLM-based multi-turn dialogue through a curiosity-driven intrinsic reward. By modeling personalization as a POMDP and rewarding the agent for improving its belief about user preferences, the approach encourages active user modeling during conversations without relying on pre-collected user data. This is a novel and valuable contribution for building personalized LLMs.  Experiments in exercise recommendation and educational dialogue show that CURIO achieves improved personalization and generalization compared to multi-turn RLHF baselines, while maintaining conversation quality. The paper is clearly written, connects well to theoretical foundations (e.g., potential-based reward shaping), and includes useful ablations and discussion of limitations.